# Generative Time Series Forecasting with Diffusion, Denoise, and Disentanglement

**Yan Li**[§ † *], **Xinjiang Lu**[† ✉], **Yaqing Wang**[†], **Dejing Dou**[†]
[†]Business Intelligence Lab, Baidu Research
[§]Zhejiang University, China
ly21121@zju.edu.cn, {luxinjiang,wangyaqing01,doudejing}@baidu.com

## Abstract

Time series forecasting has been a widely explored task of great importance in many applications. However, it is common that real-world time series data are recorded in a short time period, which results in a big gap between the deep model and the limited and noisy time series. In this work, we propose to address the time series forecasting problem with generative modeling and propose a bidirectional variational auto-encoder (BVAE) equipped with diffusion, denoise, and disentanglement, namely $D^3VAE$. Specifically, a coupled diffusion probabilistic model is proposed to augment the time series data without increasing the aleatoric uncertainty and implement a more tractable inference process with BVAE. To ensure the generated series move toward the true target, we further propose to adapt and integrate the multiscale denoising score matching into the diffusion process for time series forecasting. In addition, to enhance the interpretability and stability of the prediction, we treat the latent variable in a multivariate manner and disentangle them on top of minimizing total correlation. Extensive experiments on synthetic and real-world data show that $D^3VAE$ outperforms competitive algorithms with remarkable margins. Our implementation is available at `https://github.com/PaddlePaddle/PaddleSpatial/tree/main/research/D3VAE`.

## 1 Introduction

Time series forecasting is of great importance for risk-averse and decision-making. Traditional RNN-based methods capture temporal dependencies of the time series to predict the future. Long short-term memories (LSTMs) and gated recurrent units (GRUs) [55, 16, 15, 40] introduce the gate functions into the cell structure to handle long-term dependencies effectively. The models based on convolutional neural networks (CNNs) capture complex inner patterns of the time series through convolutional operations [28, 4, 3]. Recently, the Transformer-based models have shown great performance in time series forecasting [54, 56, 25, 29] with the help of multi-head self-attention. However, one big issue of neural networks in time series forecasting is the uncertainty [14, 1] resulting from the properties of the deep structure. The models based on vector autoregression (VAR) [5, 10, 23] try to model the distribution of time series from hidden states, which could provide more reliability to the prediction, while the performance is not satisfactory [27].

Interpretable representation learning is another merit of time series forecasting. Variational auto-encoders (VAEs) have shown not only the superiority in modeling latent distributions of the data and reducing the gradient noise [36, 24, 30, 45] but also the interpretability of time series forecasting [11, 12]. However, the interpretability of VAEs might be inferior due to the entangled latent variables.

---

[*]This work was done when the first author was an intern at Baidu Research under the supervision of the second author.

There have been efforts to learn representation disentangling [22, 2, 18], which show that the well-disentangled representation can improve the performance and robustness of the algorithm.

Moreover, real-world time series are often noisy and recorded in a short time period, which may result in overfitting and generalization issues [13, 49, 57, 41][1]. To this end, we address the time series forecasting problem with generative modeling. Specifically, we propose a bidirectional variational auto-encoder (BVAE) equipped with diffusion, denoise, and disentanglement, namely D$^3$VAE. More specifically, we first propose a coupled diffusion probabilistic model to remedy the limitation of time series data by augmenting the input time series, as well as the output time series, inspired by the forward process of the diffusion model [42, 19, 34, 35]. Besides, we adapt the Nouveau VAE [45] to the time series forecasting task and develop a BVAE as a substitute for the reverse process of the diffusion model. In this way, the expressiveness of the diffusion model plus the tractability of the VAE can be leveraged together for generative time series forecasting. Though the merit of generalizability is helpful, the diffused samples might be corrupted, which results in a generative model moving toward the noisy target. Therefore, we further develop a scaled denoising score-matching network for cleaning diffused target time series. In addition, we disentangle the latent variables of the time series by assuming that different disentangled dimensions of the latent variables correspond to different temporal patterns (such as trend, seasonality, etc.). Our contributions can be summarized as follows:

- We propose a coupled diffusion probabilistic model aiming to reduce the aleatoric uncertainty of the time series and improve the generalization capability of the generative model.
- We integrate the multiscale denoising score matching into the coupled diffusion process to improve the accuracy of generated results.
- We disentangle the latent variables of the generative model to improve the interpretability for time series forecasting.
- Extensive experiments on synthetic and real-world datasets demonstrate that D$^3$VAE outperforms competitive baselines with satisfactory margins.

## 2 Methodology

### 2.1 Generative Time Series Forecasting

**Problem Formulation.** Given an input multivariate time series $X = \{x_1, x_2, \cdots, x_n \,|\, x_i \in \mathbb{R}^d\}$ and the corresponding target time series $Y = \{y_{n+1}, y_{n+2}, \cdots, y_{n+m} \,|\, y_j \in \mathbb{R}^{d'}\}$ ($d' \leq d$). We assume that $Y$ can be generated from latent variables $Z \in \Omega_Z$ that can be drawn from the Gaussian distribution $Z \sim p(Z|X)$. The latent distribution can be further formulated as $p_\phi(Z|X) = g_\phi(X)$ where $g_\phi$ denotes a nonlinear function. Then, the data density of the target series is given by:

$$p_\theta(Y) = \int_{\Omega_Z} p_\phi(Z|X)(Y - f_\theta(Z))dZ \,, \tag{1}$$

where $f_\theta$ denotes a parameterized function. The target time series can be obtained directly by sampling from $p_\theta(Y)$.

In our problem setting, time series forecasting is to learn the representation $Z$ that captures useful signals of $X$, and map the low dimensional $X$ to the latent space with high expressiveness. The framework overview of D$^3$VAE is demonstrated in Fig. 1. Before diving into the detailed techniques, we first introduce a preliminary proposition.

**Proposition 1.** *Given a time series $X$ and its inherent noise $\epsilon_X$, we have the decomposition: $X = \langle X_r, \epsilon_X \rangle$, where $X_r$ is the ideal time series data without noise. $X_r$ and $\epsilon_X$ are independent of each other. Let $p_\phi(Z|X) = p_\phi(Z|X_r, \epsilon_X)$, the estimated target series $\widehat{Y}$ can be generated with the distribution $p_\theta(\widehat{Y}|Z) = p_\theta(\widehat{Y}_r|Z) \cdot p_\theta(\epsilon_{\widehat{Y}}|Z)$ where $\widehat{Y}_r$ is the ideal part of $\widehat{Y}$ and $\epsilon_{\widehat{Y}}$ is the estimation noise. Without loss of generality, $\widehat{Y}_r$ can be fully captured by the model. That is, $\|Y_r - \widehat{Y}_r\| \longrightarrow 0$ where $Y_r$ is the ideal part of ground truth target series $Y$. In addition, $Y$ can be decomposed as $Y = \langle \widehat{Y}_r, \epsilon_Y \rangle$ ($\epsilon_Y$ denotes the noise of $Y$). Therefore, the error between ground truth and prediction, i.e., $\|Y - \widehat{Y}\| = \|\epsilon_Y - \epsilon_{\widehat{Y}}\| > 0$, can be deemed as the combination of aleatoric uncertainty and epistemic uncertainty.*

---

[1]The detailed literature review can be found in Appendix A.

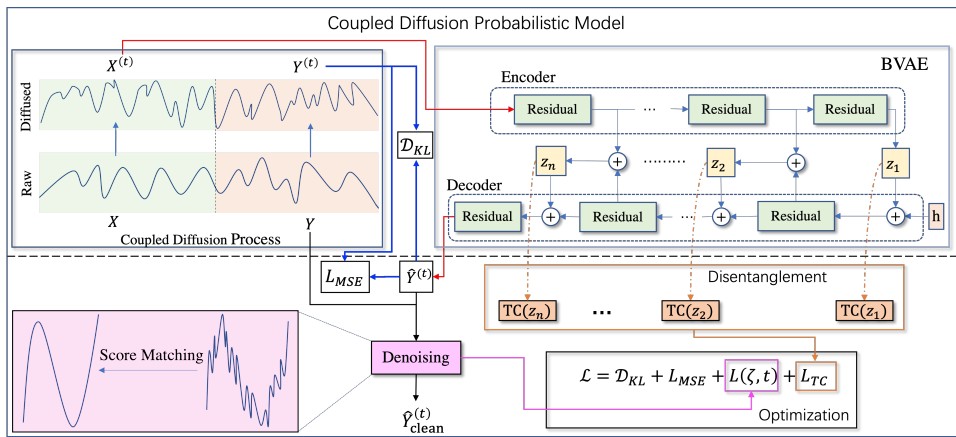

Figure 1: The framework overview of D³VAE. First, the input and output series are augmented simultaneously with the *coupled diffusion process*. Then the diffused input series are fed into a proposed BVAE model for inference, which can be deemed a *reverse process*. A denoising score-matching mechanism is applied to make the estimated target move toward the true target series. Meanwhile, the latent states in BVAE are leveraged for disentangling such that the model interpretability and reliability can be improved.

## 2.2 Coupled Diffusion Probabilistic Model

The diffusion probabilistic model (diffusion model for brevity) is a family of latent variable models aiming to generate high-quality samples. To equip the generative time series forecasting model with high expressiveness, a coupled *forward process* is developed to augment the input series and target series synchronously. Besides, in the forecasting task, more tractable and accurate prediction is expected. To achieve this, we propose a bidirectional variational auto-encoder (BVAE) to take the place of the *reverse process* in the diffusion model. We present the technical details in the following two parts, respectively.

### 2.2.1 Coupled Diffusion Process

The forward diffusion process is fixed to a Markov chain that gradually adds Gaussian noise to the data [42, 19]. To diffuse the input and output series, we propose a coupled diffusion process, which is demonstrated in Fig. 2. Specifically, given the input $X = X^{(0)} \sim q(X^{(0)})$, the approximate posterior $q(X^{(1:T)}|X^{(0)})$ can be obtained as

$$q(X^{(1:T)}|X^{(0)}) = \prod_{t=1}^{T} q(X^{(t)}|X^{(t-1)}), \quad q(X^{(t)}|X^{(t-1)}) = \mathcal{N}(X^{(t)}; \sqrt{1-\beta_t}X^{(t)}, \beta_t I), \quad (2)$$

where a uniformly increasing variance schedule $\boldsymbol{\beta} = \{\beta_1, \cdots, \beta_T \,|\, \beta_t \in [0,1)\}$ is employed to control the level of noise to be added. Then, let $\alpha_t = 1 - \beta_t$ and $\bar{\alpha}_t = \prod_{s=1}^{t} \alpha_s$, we have

$$q(X^{(t)}|X^{(0)}) = \mathcal{N}(X^{(t)}; \sqrt{\bar{\alpha}_t}X^{(0)}, (1-\bar{\alpha}_t)I). \quad (3)$$

Furthermore, according to Proposition 1 we decompose $X^{(0)}$ as $X^{(0)} = \langle X_r, \epsilon_X \rangle$. Then, with Eq. (3), the diffused $X^{(t)}$ can be decomposed as follows:

$$X^{(t)} = \sqrt{\bar{\alpha}_t}X^{(0)} + (1-\bar{\alpha}_t)\delta_X := \langle \underbrace{\sqrt{\bar{\alpha}_t}X_r}_{\text{ideal part}}, \underbrace{\sqrt{\bar{\alpha}_t}\epsilon_X + (1-\bar{\alpha}_t)\delta_X}_{\text{noisy part}} \rangle, \quad (4)$$

where $\delta_X$ denotes the standard Gaussian noise of $X$. As $\boldsymbol{\alpha}$ can be determined when the variance schedule $\boldsymbol{\beta}$ is known, the ideal part is also determined in the diffusion process. Let $\widetilde{X}_r^{(t)} = \sqrt{\bar{\alpha}_t}X_r$ and $\delta_{\widetilde{X}}^{(t)} = \sqrt{\bar{\alpha}_t}\epsilon_X + (1-\bar{\alpha}_t)\delta_X$, then, according to Proposition 1 and Eq. (4), we have

$$p_\phi(Z^{(t)}|X^{(t)}) = p_\phi(Z^{(t)}|\widetilde{X}_r^{(t)}, \delta_{\widetilde{X}}^{(t)}), \quad p_\theta(\widehat{Y}^{(t)}|Z^{(t)}) = p_\theta(\widehat{Y}_r^{(t)}|Z^{(t)})p_\theta(\delta_{\widehat{Y}}^{(t)}|Z^{(t)}), \quad (5)$$

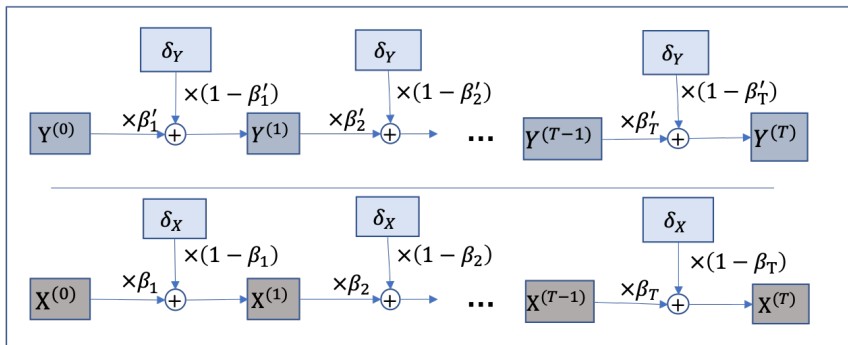

Figure 2: An illustration of the coupled diffusion process. The input $X^{(0)}$ and the corresponding target $Y^{(0)}$ are diffused simultaneously with different variance schedules. $\boldsymbol{\beta} = \{\beta_1, \cdots, \beta_T\}$ is the variance schedule for the input and $\boldsymbol{\beta'} = \{\beta'_1, \cdots, \beta'_T\}$ is for the target.

where $\delta_{\widehat{Y}}^{(t)}$ denotes the generated noise of $\widehat{Y}^{(t)}$. To relieve the effect of aleatoric uncertainty resulting from time series data, we further apply the diffusion process to the target series $Y = Y^{(0)} \sim q(Y^{(0)})$. In particular, a scale parameter $\omega \in (0,1)$ is adopted, such that $\beta'_t = \omega\beta_t, \alpha'_t = 1 - \beta'_t$ and $\bar{\alpha}'_t = \prod_{s=1}^{t} \alpha'_s$. Then, according to Proposition 1, we can obtain the following decomposition (similar to Eq. (4)):

$$Y^{(t)} = \sqrt{\bar{\alpha}'_t}Y^{(0)} + (1 - \bar{\alpha}'_t)\delta_Y := \langle \underbrace{\sqrt{\bar{\alpha}'_t}Y_r}_{\text{ideal part}}, \underbrace{\sqrt{\bar{\alpha}'_t}\epsilon_Y + (1 - \bar{\alpha}'_t)\delta_Y}_{\text{noisy part}} \rangle = \langle \widetilde{Y}_r^{(t)}, \delta_{\widetilde{Y}}^{(t)} \rangle. \quad (6)$$

Consequently, we have $q(Y^{(t)}) = q(\widetilde{Y}_r^{(t)})q(\delta_{\widetilde{Y}}^{(t)})$. Afterward, we can draw the following conclusions with Proposition 1 and Eqs. (5) and (6). The proofs can be found in Appendix B.

**Lemma 1.** $\forall \varepsilon > 0$, *there exists a probabilistic model* $f_{\phi,\theta} := (p_\phi, p_\theta)$ *to guarantee that* $\mathcal{D}_{\text{KL}}(q(\widetilde{Y}_r^{(t)})||p_\theta(\widehat{Y}_r^{(t)})) < \varepsilon$, *where* $\widehat{Y}_r^{(t)} = f_{\phi,\theta}(X^{(t)})$.

**Lemma 2.** *With the coupled diffusion process, the difference between diffusion noise and generation noise will be reduced, i.e.,* $\lim_{t\to\infty} \mathcal{D}_{\text{KL}}(q(\delta_{\widehat{Y}}^{(t)})||p_\theta(\delta_{\widehat{Y}}^{(t)}|Z^{(t)})) < \mathcal{D}_{\text{KL}}(q(\epsilon_Y)||p_\theta(\epsilon_{\widehat{Y}}))$ .

Therefore, the uncertainty raised by the generative model and the inherent data noise can be reduced through the coupled diffusion process. In addition, the diffusion process simultaneously augments the input series and the target series, which can improve the generalization capability for (esp. short) time series forecasting.

### 2.2.2 Bidirectional Variational Auto-Encoder

Traditionally, in the diffusion model, a reverse process is adopted to generate high-quality samples [42, 19]. However, for the generative time series forecasting problem, not only the expressiveness but also the supervision of the ground truths should be considered. In this work, we employ a more efficient generative model, i.e., bidirectional variational auto-encoder (BVAE) [45], to take the place of the reverse process of the diffusion model. The architecture of BVAE is described in Fig. 1 where $Z$ is treated in a multivariate fashion $Z = \{z_1, \cdots, z_n\}$ ($z_i \in \mathbb{R}^m, z_i = [z_{i,1}, \cdots, z_{i,m}]$) and $z_{i+1} \sim p(z_{i+1}|z_i, X)$. Then, $n$ is determined in accordance with the number of residual blocks in the encoder, as well as the decoder. Another merit of BVAE is that it opens an interface to integrate the disentanglement for improving model interpretability (refer to Section 2.4).

### 2.3 Scaled Denoising Score Matching for Diffused Time Series Cleaning

Although the time series data can be augmented with the aforementioned coupled diffusion probabilistic model, the generative distribution $p_\theta(\widehat{Y}^{(t)})$ tends to move toward the diffused target series $Y^{(t)}$ which has been corrupted [32, 43]. To further "clean" the generated target series, we employ the Denoising Score Matching (DSM) to accelerate the de-uncertainty process without sacrificing the

model flexibility. DSM [46, 32] was proposed to link Denoising Auto-Encoder (DAE) [47] to Score Matching (SM) [20]. Let $\widehat{Y}$ denote the generated target series, then we have the objective

$$L_{\text{DSM}}(\zeta) = \mathbb{E}_{p_{\sigma_0}(\widehat{Y},Y)}\|\nabla_{\widehat{Y}}\log(q_{\sigma_0}(\widehat{Y}|Y)) + \nabla_{\widehat{Y}}E(\widehat{Y};\zeta)\|^2\,, \tag{7}$$

where $p_{\sigma_0}(\widehat{Y}, Y)$ is the joint density of pairs of corrupted and clean samples $(\widehat{Y}, Y)$, $\nabla_{\widehat{Y}}\log(q_{\sigma_0}(\widehat{Y}|Y))$ is derivative of the log density of a single noise kernel, which is dedicated to replacing the Parzen density estimator: $p_{\sigma_0}(\widehat{Y}) = \int q_{\sigma_0}(\widehat{Y}|Y)p(Y)dY$ in score matching, and $E(\widehat{Y};\zeta)$ is the energy function. In the particular case of Gaussian noise, $\log(q_{\sigma_0}(\widehat{Y}|Y)) = -(\widehat{Y}-Y)^2/2\sigma_0^2 + C$. Thus, we have

$$L_{\text{DSM}}(\zeta) = \mathbb{E}_{p_{\sigma_0}(\widehat{Y},Y)}\|Y - \widehat{Y} + \sigma_0^2\nabla_{\widehat{Y}}E(\widehat{Y};\zeta)\|^2\,. \tag{8}$$

Then, for the diffused target series at step $t$, we can obtain

$$L_{\text{DSM}}(\zeta, t) = \mathbb{E}_{p_{\sigma_0}(\widehat{Y}^{(t)},Y)}\|Y - \widehat{Y}^{(t)} + \sigma_0^2\nabla_{\widehat{Y}^{(t)}}E(\widehat{Y}^{(t)};\zeta)\|^2\,. \tag{9}$$

To scale the noise of different levels [32], a monotonically decreasing series of fixed $\sigma$ values $\{\sigma_1,\cdots,\sigma_T \,|\, \sigma_t = 1 - \bar{\alpha}_t\}$ (refer to the aforementioned variance schedule $\boldsymbol{\beta}$ in Section 2.2) is adopted. Therefore, the objective of the multi-scaled DSM is

$$L(\zeta, t) = \mathbb{E}_{q_\sigma(\widehat{Y}^{(t)}|Y)p(Y)}l(\sigma_t)\|Y - \widehat{Y}^{(t)} + \sigma_0^2\nabla_{\widehat{Y}^{(t)}}E(\widehat{Y}^{(t)};\zeta)\|^2\,, \tag{10}$$

where $\sigma \in \{\sigma_1,\cdots,\sigma_T\}$ and $l(\sigma_t) = \sigma_t$. With Eq. (10), we can ensure that the gradient has the right magnitude by setting $\sigma_0$.

In the generative time series forecasting setting, the generated samples will be tested without applying the diffusion process. To further denoise the generated target series $\widehat{Y}$, we apply a single-step gradient denoising jump [39]:

$$\widehat{Y}_{\text{clean}} = \widehat{Y} - \sigma_0^2\nabla_{\widehat{Y}}E(\widehat{Y};\zeta)\,. \tag{11}$$

The generated results tend to possess a larger distribution space than the true target, and the noisy term in Eq. (11) approximates the noise between the generated target series and the "cleaned" target series. Therefore, $\sigma_0^2\nabla_{\widehat{Y}}E(\widehat{Y};\zeta)$ can be treated as the estimated uncertainty of the prediction.

## 2.4 Disentangling Latent Variables for Interpretation

The interpretability of the time series forecasting model is of great importance for many downstream tasks [44, 17, 21]. Through disentangling the latent variables of the generative model, not only the interpretability but also the reliability of the prediction can be further enhanced [31].

To disentangle the latent variables $Z = \{z_1,\cdots,z_n\}$, we attempt to minimize the Total Correlation (TC) [50, 22], which is a popular metric to measure dependencies among multiple random variables,

$$\text{TC}(z_i) = \mathcal{D}_{\text{KL}}(p_\phi(z_i)||\bar{p}_\phi(z_i)), \qquad \bar{p}_\phi(z_i) = \prod_{j=1}^{m} p_\phi(z_{i,j}) \tag{12}$$

where $m$ denotes the number of factors of $z_i$ that need to be disentangled. Lower TC generally means better disentanglement if the latent variables preserve useful information. However, a very low TC can still be obtained when the latent variables carry no meaningful signals. Through the bidirectional structure of BVAE, such issues can be tackled without too much effort. As shown in Fig. 1, the signals are disseminated in both the encoder and decoder, such that rich semantics are aggregated into the latent variables. Furthermore, to alleviate the effect of potential irregular values, we average the total correlations of $z_{1:n}$, then the loss w.r.t. the TC score of BVAE can be obtained:

$$L_{\text{TC}} = \frac{1}{n}\sum_{i=1}^{n} \text{TC}(z_i)\,. \tag{13}$$

---

**Algorithm 1** Training Procedure.

---

1: **repeat**
2:     $X^{(0)} \sim q(X^{(0)}), \quad Y^{(0)} \sim q(Y^{(0)}), \quad \delta_X \sim N(0, I_d), \quad \delta_Y \sim N(0, I_d)$
3:     Randomly choose $t \in \{1, \cdots, T\}$ and with Eqs. (4) and (6),
4:       $X^{(t)} = \sqrt{\bar{\alpha}_t} X^{(0)} + (1 - \bar{\alpha}_t) \delta_X, \quad Y^{(t)} = \sqrt{\bar{\alpha}'_t} Y^{(0)} + (1 - \bar{\alpha}'_t) \delta_Y$
5:     Generate the latent variable $Z$ with BVAE, $Z \sim p_\phi(Z|X^{(t)})$
6:     Sample $\widehat{Y}^{(t)} \sim p_\theta(\widehat{Y}^{(t)}|Z)$ and calculate $\mathcal{D}_{\mathrm{KL}}(q(Y^{(t)})||p_\theta(\widehat{Y}^{(t)}))$
7:     Calculate DSM loss with Eq. (10)
8:     Calculate total correlation of $Z$ with Eq. (13)
9:     Construct the total loss $\mathcal{L}$ with Eq. (14)
10:     $\theta, \phi \leftarrow \mathrm{argmin}(\mathcal{L})$
11: **until** Convergence

---

---

**Algorithm 2** Forecasting Procedure.

---

1: **Input:** $X \sim q(X)$
2: Sample $Z \sim p_\phi(Z|X)$
3: Generate $\widehat{Y} \sim p_\theta(\widehat{Y}|Z)$
4: **Output:** $\widehat{Y}_{\mathrm{clean}}$ and the estimated uncertainty with Eq. (11)

---

## 2.5 Training and Forecasting

**Training Objective.** To reduce the effect of uncertainty, the coupled diffusion equipped with the denoising network is proposed without sacrificing generalizability. Then we disentangle the latent variables of the generative model by minimizing the TC of the latent variables. Finally, we reconstruct the loss with several trade-off parameters, and with Eqs. (10), (11) and (13) we have

$$\mathcal{L} = \psi \cdot \mathcal{D}_{\mathrm{KL}}(q(Y^{(t)})||p_\theta(\widehat{Y}^{(t)})) + \lambda \cdot L(\zeta, t) + \gamma \cdot L_{\mathrm{TC}} + L_{\mathrm{mse}}(\widehat{Y}^{(t)}, Y^{(t)}), \tag{14}$$

where $L_{\mathrm{mse}}$ calculates the mean square error (MSE) between $\widehat{Y}^{(t)}$ and $Y^{(t)}$. We minimize the above objective to learn the generative model accordingly.

**Algorithms.** Algorithm 1 displays the complete training procedure of D³VAE with the loss function in Eq. (14). For inference, as described in Algorithm 2, given the input series $X$, the target series can be generated directly from the distribution $p_\theta$ which is conditioned on the latent states drawn from the distribution $p_\phi$.

# 3 Experiments

## 3.1 Experiment Settings

**Datasets.** We generate two synthetic datasets suggested by [9],

$$w_t = a \cdot w_{t-1} + \tanh(b \cdot w_{t-2}) + \sin(w_{t-3}) + \mathcal{N}(0, 0.5I)$$
$$X = [w_1, w_2, ..., w_N] \cdot F + \mathcal{N}(0, 0.5I),$$

where $w_t \in \mathbb{R}^2$ and $0 \le w_{t,1}, w_{t,2} \le 1$ $(t = 1, 2, 3)$, $F \in \mathbb{R}^{2 \times k} \sim \mathcal{U}[-1, 1]$, $k$ denotes the dimensionality and $N$ is the number of time points, $a, b$ are two constants. We set $a = 0.9, b = 0.2, k = 20$ to generate $\mathrm{D}_1$, and $a = 0.5, b = 0.5, k = 40$ for $\mathrm{D}_2$, and $N = 800$ for both $\mathrm{D}_1$ and $\mathrm{D}_2$.

Six real-world datasets with diverse spatiotemporal dynamics are selected, including Traffic [27], Electricity[2], Weather[3], Wind (Wind Power) [4], and ETTs [56] (ETTm1 and ETTh1). To highlight the uncertainty in short time series scenarios, for each dataset, we slice a subset from the starting point to make sure that each sliced dataset contains at most 1000 time points. Subsequently, we

---

[2]`https://archive.ics.uci.edu/ml/datasets/ElectricityLoadDiagrams20112014`
[3]`https://www.bgc-jena.mpg.de/wetter/`
[4]This dataset is published at `https://github.com/PaddlePaddle/PaddleSpatial/tree/main/paddlespatial/datasets/WindPower`.

Table 1: Performance comparisons on synthetic data in terms of MSE and CRPS. The best results are boldfaced.

| Model | | $D^3$VAE | NVAE | $\beta$-TCVAE | f-VAE | DeepAR | TimeGrad | GP-copula | VAE |
|---|---|---|---|---|---|---|---|---|---|
| $D_1$ | 8 | **0.512**$_{\pm.033}$ | 1.201$_{\pm.027}$ | 0.631$_{\pm.003}$ | 0.854$_{\pm.099}$ | 1.153$_{\pm.125}$ | 0.966$_{\pm.102}$ | 1.202$_{\pm.108}$ | 0.912$_{\pm.132}$ |
| | | **0.585**$_{\pm.021}$ | 0.905$_{\pm.011}$ | 0.658$_{\pm.002}$ | 0.745$_{\pm.036}$ | 0.758$_{\pm.038}$ | 0.698$_{\pm.024}$ | 0.773$_{\pm.033}$ | 0.786$_{\pm.053}$ |
| | 16 | **0.571**$_{\pm.025}$ | 1.184$_{\pm.025}$ | 0.758$_{\pm.047}$ | 1.046$_{\pm.270}$ | 0.911$_{\pm.046}$ | 0.945$_{\pm.315}$ | 0.915$_{\pm.059}$ | 0.908$_{\pm.177}$ |
| | | **0.625**$_{\pm.013}$ | 0.897$_{\pm.012}$ | 0.747$_{\pm.027}$ | 0.835$_{\pm.108}$ | 0.699$_{\pm.014}$ | 0.709$_{\pm.100}$ | 0.704$_{\pm.020}$ | 0.765$_{\pm.067}$ |
| $D_2$ | 8 | **0.599**$_{\pm.049}$ | 1.966$_{\pm.047}$ | 3.096$_{\pm.197}$ | 3.353$_{\pm.430}$ | 0.977$_{\pm.137}$ | 0.963$_{\pm.385}$ | 1.037$_{\pm.082}$ | 3.079$_{\pm.345}$ |
| | | **0.628**$_{\pm.027}$ | 1.255$_{\pm.021}$ | 1.680$_{\pm.062}$ | 1.640$_{\pm.154}$ | 0.727$_{\pm.058}$ | 0.706$_{\pm.123}$ | 0.753$_{\pm.026}$ | 1.504$_{\pm.098}$ |
| | 16 | **0.786**$_{\pm.041}$ | 1.955$_{\pm.051}$ | 3.067$_{\pm.443}$ | 3.109$_{\pm.428}$ | 0.972$_{\pm.144}$ | 0.850$_{\pm.061}$ | 1.082$_{\pm.071}$ | 3.132$_{\pm.160}$ |
| | | 0.728$_{\pm.026}$ | 1.251$_{\pm.020}$ | 1.643$_{\pm.183}$ | 1.558$_{\pm.157}$ | 0.720$_{\pm.050}$ | **0.649**$_{\pm.017}$ | 0.762$_{\pm.008}$ | 1.560$_{\pm.060}$ |

Table 2: The performance comparisons on real-world datasets in terms of MSE and CRPS, and the best results are in boldface.

| Model | | $D^3$VAE | NVAE | $\beta$-TCVAE | f-VAE | DeepAR | TimeGrad | GP-copula | VAE |
|---|---|---|---|---|---|---|---|---|---|
| Traffic | 8 | **0.081**$_{\pm.003}$ | 1.300$_{\pm.024}$ | 1.003$_{\pm.006}$ | 0.982$_{\pm.059}$ | 3.895$_{\pm.306}$ | 3.695$_{\pm.246}$ | 4.299$_{\pm.372}$ | 0.794$_{\pm.130}$ |
| | | **0.207**$_{\pm.003}$ | 0.593$_{\pm.004}$ | 0.894$_{\pm.003}$ | 0.666$_{\pm.032}$ | 1.391$_{\pm.071}$ | 1.410$_{\pm.027}$ | 1.408$_{\pm.046}$ | 0.759$_{\pm.07}$ |
| | 16 | **0.081**$_{\pm.009}$ | 1.271$_{\pm.019}$ | 0.997$_{\pm.004}$ | 0.998$_{\pm.042}$ | 4.141$_{\pm.320}$ | 3.495$_{\pm.362}$ | 4.575$_{\pm.141}$ | 0.632$_{\pm.057}$ |
| | | **0.200**$_{\pm.014}$ | 0.589$_{\pm.001}$ | 0.893$_{\pm.002}$ | 0.692$_{\pm.026}$ | 1.338$_{\pm.043}$ | 1.329$_{\pm.057}$ | 1.506$_{\pm.025}$ | 0.671$_{\pm.038}$ |
| Electricity | 8 | **0.251**$_{\pm.015}$ | 1.134$_{\pm.029}$ | 0.901$_{\pm.052}$ | 0.893$_{\pm.069}$ | 2.934$_{\pm.173}$ | 2.703$_{\pm.087}$ | 2.924$_{\pm.218}$ | 0.853$_{\pm.040}$ |
| | | **0.398**$_{\pm.011}$ | 0.542$_{\pm.003}$ | 0.831$_{\pm.004}$ | 0.809$_{\pm.024}$ | 1.244$_{\pm.037}$ | 1.208$_{\pm.024}$ | 1.249$_{\pm.048}$ | 0.795$_{\pm.016}$ |
| | 16 | **0.308**$_{\pm.030}$ | 1.150$_{\pm.032}$ | 0.850$_{\pm.003}$ | 0.807$_{\pm.034}$ | 2.803$_{\pm.199}$ | 2.770$_{\pm.237}$ | 3.065$_{\pm.186}$ | 0.846$_{\pm.062}$ |
| | | **0.437**$_{\pm.020}$ | 0.531$_{\pm.003}$ | 0.814$_{\pm.002}$ | 0.782$_{\pm.024}$ | 1.220$_{\pm.048}$ | 1.240$_{\pm.048}$ | 1.307$_{\pm.042}$ | 0.793$_{\pm.029}$ |
| Weather | 8 | **0.169**$_{\pm.022}$ | 0.801$_{\pm.024}$ | 0.234$_{\pm.042}$ | 0.591$_{\pm.198}$ | 2.317$_{\pm.357}$ | 2.715$_{\pm.189}$ | 2.412$_{\pm.761}$ | 0.560$_{\pm.192}$ |
| | | **0.357**$_{\pm.024}$ | 0.757$_{\pm.013}$ | 0.404$_{\pm.040}$ | 0.565$_{\pm.080}$ | 0.858$_{\pm.078}$ | 0.920$_{\pm.013}$ | 0.897$_{\pm.115}$ | 0.572$_{\pm.077}$ |
| | 16 | **0.187**$_{\pm.047}$ | 0.811$_{\pm.016}$ | 0.212$_{\pm.012}$ | 0.530$_{\pm.167}$ | 1.269$_{\pm.187}$ | 1.110$_{\pm.083}$ | 1.357$_{\pm.145}$ | 0.424$_{\pm.141}$ |
| | | **0.361**$_{\pm.046}$ | 0.759$_{\pm.009}$ | 0.388$_{\pm.014}$ | 0.547$_{\pm.067}$ | 0.783$_{\pm.059}$ | 0.733$_{\pm.016}$ | 0.811$_{\pm.032}$ | 0.503$_{\pm.068}$ |
| ETTm1 | 8 | **0.527**$_{\pm.073}$ | 0.921$_{\pm.026}$ | 1.538$_{\pm.254}$ | 2.326$_{\pm.445}$ | 2.204$_{\pm.420}$ | 1.877$_{\pm.245}$ | 2.024$_{\pm.143}$ | 2.375$_{\pm.405}$ |
| | | **0.557**$_{0.048}$ | 0.760$_{\pm.026}$ | 1.015$_{\pm.112}$ | 1.260$_{\pm.167}$ | 0.984$_{\pm.074}$ | 0.908$_{\pm.038}$ | 0.961$_{\pm.027}$ | 1.258$_{\pm.104}$ |
| | 16 | **0.968**$_{\pm.104}$ | 1.100$_{\pm.032}$ | 1.744$_{\pm.100}$ | 2.339$_{\pm.270}$ | 2.350$_{\pm.170}$ | 2.032$_{\pm.234}$ | 2.486$_{\pm.207}$ | 2.321$_{\pm.469}$ |
| | | **0.821**$_{\pm.072}$ | 0.822$_{\pm.026}$ | 1.104$_{\pm.041}$ | 1.249$_{\pm.088}$ | 0.974$_{\pm.016}$ | 0.919$_{\pm.031}$ | 0.984$_{\pm.016}$ | 1.259$_{\pm.132}$ |
| ETTh1 | 8 | **0.292**$_{\pm.036}$ | 0.483$_{\pm.017}$ | 0.703$_{\pm.054}$ | 0.870$_{\pm.134}$ | 3.451$_{\pm.335}$ | 4.259$_{\pm1.13}$ | 4.278$_{\pm1.12}$ | 1.006$_{\pm.281}$ |
| | | **0.424**$_{\pm.033}$ | 0.461$_{\pm.011}$ | 0.644$_{\pm.038}$ | 0.730$_{\pm.060}$ | 1.194$_{\pm.034}$ | 1.092$_{\pm.028}$ | 1.169$_{\pm.055}$ | 0.762$_{\pm.115}$ |
| | 16 | **0.374**$_{\pm.061}$ | 0.488$_{\pm.010}$ | 0.681$_{\pm.018}$ | 0.983$_{\pm.139}$ | 1.929$_{\pm.105}$ | 1.332$_{\pm.125}$ | 1.701$_{\pm.088}$ | 0.681$_{\pm.104}$ |
| | | 0.488$_{\pm.039}$ | **0.463**$_{\pm.018}$ | 0.640$_{\pm.008}$ | 0.760$_{\pm.062}$ | 1.029$_{\pm.030}$ | 0.879$_{\pm.037}$ | 0.999$_{\pm.023}$ | 0.641$_{\pm.055}$ |
| Wind | 8 | **0.681**$_{\pm.075}$ | 1.854$_{\pm.032}$ | 1.321$_{\pm.379}$ | 1.942$_{\pm.101}$ | 12.53$_{\pm2.25}$ | 12.67$_{\pm1.75}$ | 11.35$_{\pm6.61}$ | 2.006$_{\pm.145}$ |
| | | **0.596**$_{\pm.052}$ | 1.223$_{\pm.014}$ | 0.863$_{\pm.143}$ | 1.067$_{\pm.086}$ | 1.370$_{\pm.107}$ | 1.440$_{\pm.059}$ | 1.305$_{\pm.369}$ | 1.103$_{\pm.100}$ |
| | 16 | 1.033$_{\pm.062}$ | 1.955$_{\pm.015}$ | **0.894**$_{\pm.038}$ | 1.262$_{\pm.178}$ | 13.96$_{\pm.1.53}$ | 12.86$_{\pm2.60}$ | 13.79$_{\pm5.37}$ | 1.138$_{\pm.205}$ |
| | | **0.757**$_{\pm.053}$ | 1.247$_{\pm.011}$ | 0.785$_{\pm.037}$ | 0.843$_{\pm.066}$ | 1.347$_{\pm.060}$ | 1.240$_{\pm.070}$ | 1.261$_{\pm.171}$ | 0.862$_{\pm.092}$ |

obtained 5%-Traffic, 3%-Electricity, 2%-Weather, 2%-Wind, 1%-ETTm1, and 5%-ETTh1. The statistical descriptions of the real-world datasets can be found in Appendix C.1. All datasets are split chronologically and adopt the same train/validation/test ratios, i.e., 7:1:2.

**Baselines.** We compare $D^3$VAE with one GP (Gaussian Process) based method (GP-copula [37]), two auto-regressive methods (DeepAR [38] and TimeGrad [35]), and four VAE-based methods, i.e., vanilla VAE, NVAE [45], factor-VAE (f-VAE for short) [22] and $\beta$-TCVAE [6].

**Implementation Details.** An input-$l_x$-predict-$l_y$ window is applied to roll the train, validation, and test sets with stride one time-step, respectively, and this setting is adopted for all datasets. Hereinafter, the last dimension of the multivariate time series is selected as the target variable by default.

We use the Adam optimizer with an initial learning rate of $5e - 4$. The batch size is 16, and the training is set to 20 epochs at most equipped with early stopping. The number of disentanglement factors is chosen from $\{4, 8\}$, and $\beta_t \in \boldsymbol{\beta}$ is set to range from $0.01$ to $0.1$ with different diffusion steps $T \in [100, 1000]$, then $\omega$ is set to $0.1$. The trade-off hyperparameters are set as $\psi = 0.05, \lambda = 0.1, \gamma = 0.001$ for ETTs, and $\psi = 0.5, \lambda = 1.0, \gamma = 0.01$ for others. All the experiments were carried out on a Linux machine with a single NVIDIA P40 GPU. The experiments are repeated five times, and the average and variance of the predictions are reported. We use the Continuous Ranked Probability Score (CRPS) [33] and Mean Squared Error (MSE) as the evaluation metrics. For both metrics, the lower, the better. In particular, CRPS is used to evaluate the similarity of two distributions and is equivalent to Mean Absolute Error (MAE) when two distributions are discrete.

## 3.2 Main Results

Two different prediction lengths, i.e., $l_y \in \{8, 16\}$ ($l_x = l_y$), are evaluated. The results of longer prediction lengths are available in Appendix D.

**Toy Datasets.** In Table 1, we can observe that D$^3$VAE achieves SOTA performance most of the time, and achieves competitive CRPS in D$_2$ for prediction length 16. Besides, VAEs outperform VARs and GP on D$_1$, but VARs achieve better performance on D$_2$, which demonstrates the advantage of VARs in learning complex temporal dependencies.

**Real-World Datasets.** As for the experiments on real-world data, D$^3$VAE achieves consistent SOTA performance except for the prediction length 16 on the Wind dataset (Table 2). Particularly, under the input-8-predict-8 setting, D$^3$VAE can provide remarkable improvements in Traffic, Electricity, Wind, ETTm1, ETTh1 and Weather w.r.t. MSE reduction (90%, 71%, 48%, 43%, 40% and 28%). Regarding the CRPS reduction, D$^3$VAE achieves a 73% reduction in Traffic, 31% in Wind, and 27% in Electricity under the input-8-predict-8 setting, and a 70% reduction in Traffic, 18% in Electricity, and 7% in Weather under the input-16-predict-16 setting. Overall, D$^3$VAE gains the averaged 43% MSE reduction and 23% CRPS reduction among the above settings. More results under longer prediction-length settings and on full datasets can be found in Appendix D.1.

**Uncertainty Estimation.** The uncertainty can be assessed by estimating the noise of the outcome series when doing the prediction (see Section 2.3). Through scale parameter $\omega$, the generated distribution space can be adjusted accordingly (results on the effect of $\omega$ can be found in Appendix D.3). The showcases in Fig. 3 demonstrate the uncertainty estimation of the yielded series in the Traffic dataset, where the last six dimensions are treated as target variables. We can find that noise estimation can quantify the uncertainty effectively. For example, the estimated uncertainty grows rapidly when extreme values are encountered.

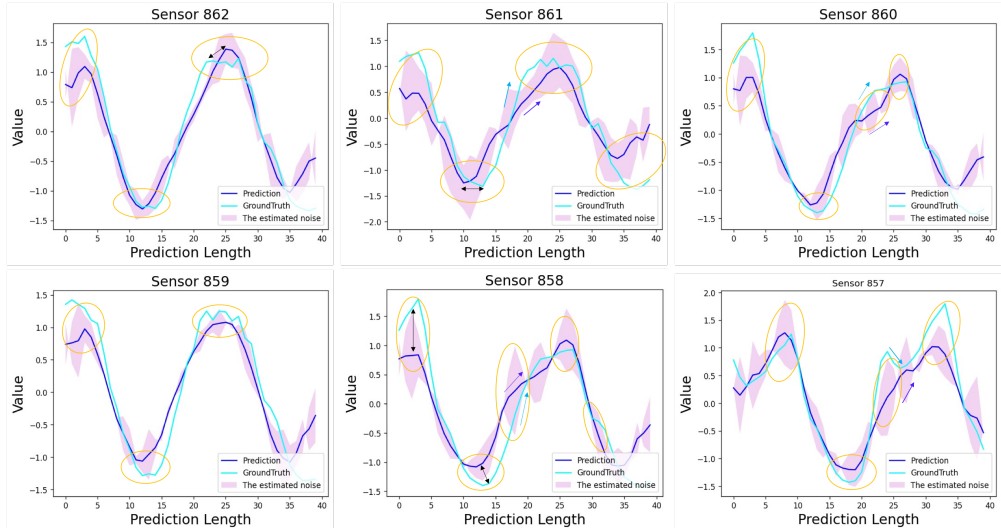

Figure 3: Uncertainty estimation of the prediction of the last six dimensions in the Traffic dataset and the colored envelope denotes the estimated uncertainty.

Table 3: Ablation study of the coupled diffusion probabilistic model w.r.t. MSE and CSPR.

| Dataset | Traffic | | Electricity | |
|---|---|---|---|---|
| | 16 | 32 | 16 | 32 |
| $\text{D}^3\text{VAE}_{-\widetilde{Y}}$ | $0.122_{\pm.006}$ $0.250_{\pm.008}$ | $0.126_{\pm.013}$ $0.261_{\pm.017}$ | $0.350_{\pm.043}$ $0.480_{\pm.032}$ | $0.422_{\pm.012}$ $0.551_{\pm.012}$ |
| $\text{D}^3\text{VAE}_{-\widetilde{Y}-\text{DSM}}$ | $0.096_{\pm.006}$ $0.217_{\pm.010}$ | $0.092_{\pm.008}$ $0.220_{\pm.013}$ | $0.331_{\pm.023}$ $0.450_{\pm.021}$ | $0.502_{\pm.079}$ $0.584_{\pm.053}$ |
| $\text{D}^3\text{VAE}_{-\widetilde{X}}$ | $0.123_{\pm.003}$ $0.256_{\pm.006}$ | $0.117_{\pm.007}$ $0.253_{\pm.013}$ | $0.351_{\pm.047}$ $0.481_{\pm.036}$ | $0.420_{\pm.056}$ $0.540_{\pm.046}$ |
| $\text{D}^3\text{VAE}_{-\text{CDM}}$ | $0.123_{\pm.004}$ $0.255_{\pm.007}$ | $0.118_{\pm.008}$ $0.252_{\pm.015}$ | $0.365_{\pm.025}$ $0.498_{\pm.018}$ | $0.439_{\pm.014}$ $0.561_{\pm.016}$ |
| $\text{D}^3\text{VAE}_{-\text{CDM}-\text{DSM}}$ | $0.123_{\pm.003}$ $0.255_{\pm.003}$ | $0.119_{\pm.003}$ $0.253_{\pm.005}$ | $0.338_{\pm.041}$ $0.467_{\pm.029}$ | $0.448_{\pm.062}$ $0.555_{\pm.041}$ |
| $\text{D}^3\text{VAE}$ | $\mathbf{0.081}_{\pm.009}$ $\mathbf{0.200}_{\pm.014}$ | $\mathbf{0.091}_{\pm.007}$ $\mathbf{0.216}_{\pm.012}$ | $\mathbf{0.308}_{\pm.030}$ $\mathbf{0.437}_{\pm.020}$ | $\mathbf{0.410}_{\pm.075}$ $\mathbf{0.534}_{\pm.058}$ |

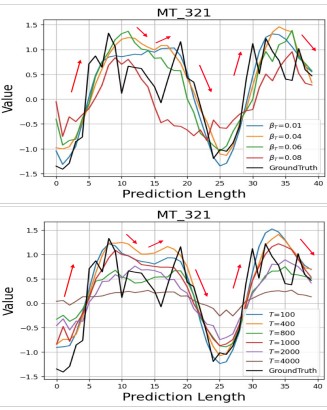

Figure 4: Comparisons of predictions with different $\beta_T$ and varying $T$ on the Electricity dataset.

**Disentanglement Evaluation.** For time series forecasting, it is difficult to label disentangled factors by hand, thus we take different dimensions of $Z$ as the factors to be disentangled: $z_i = [z_{i,1}, \cdots, z_{i,m}]$ ($z_i \in Z$). We build a classifier to discriminate whether an instance $z_{i,j}$ belongs to class $j$ such that the disentanglement quality can be assessed by evaluating the classification performance. Besides, we adopt the Mutual Information Gap (MIG) [6] as a metric to evaluate the disentanglement more straightforwardly. Due to the space limit, the evaluation of disentanglement with different factors can be found in Appendix E.

### 3.3 Model Analysis

**Ablation Study of the Coupled Diffusion and Denoising Network.** To evaluate the effectiveness of the coupled diffusion model (CDM), we compare the full versioned $\text{D}^3\text{VAE}$ with its three variants: i) $\text{D}^3\text{VAE}_{-\widetilde{Y}}$, i.e. $\text{D}^3\text{VAE}$ without diffused $Y$, ii) $\text{D}^3\text{VAE}_{-\widetilde{X}}$, i.e. $\text{D}^3\text{VAE}$ without diffused $X$, and iii) $\text{D}^3\text{VAE}_{-\text{CDM}}$, i.e. $\text{D}^3\text{VAE}$ without any diffusion. Besides, the performance of $\text{D}^3\text{VAE}$ without denoising score matching (DSM) is also reported when the target series is not diffused, which are denoted as $\text{D}^3\text{VAE}_{-\widetilde{Y}-\text{DSM}}$ and $\text{D}^3\text{VAE}_{-\text{CDM}-\text{DSM}}$. The ablation study is carried out on Traffic and Electricity datasets under input-16-predict-16 and input-32-predict-32. In Table 3, we can find that the diffusion process can effectively augment the input or the target. Moreover, when the target is not diffused, the denoising network would be deficient since the noise level of the target cannot be estimated by then.

**Variance Schedule $\beta$ and The Number of Diffusion Steps $T$.** To reduce the effect of the uncertainty while preserving the informative temporal patterns, the extent of the diffusion should be configured properly. Too small a variance schedule or inadequate diffusion steps will lead to a meaningless diffusion process. Otherwise, the diffusion could be out of control [5]. Here we analyze the effect of the variance schedule $\beta$ and the number of diffusion steps $T$. We set $\beta_1 = 0$ and change the value of $\beta_t$ in the range of $[0.01, 0.1]$, and $T$ ranges from 100 to 4000. As shown in Fig. 4, we can see that the prediction performance can be improved if proper $\beta$ and $T$ are employed.

## 4  Discussion

**Sampling for Generative Time Series Forecasting.**
The Langevin dynamics has been widely applied to the sampling of energy-based models (EBMs) [51, 8, 53],

$$Y_k = Y_{k-1} - \frac{\rho}{2}\nabla_Y E_\phi(Y_{k-1}) + \rho^{\frac{1}{2}}\mathcal{N}(0, I_d)\,, \tag{15}$$

---

[5]An illustrative showcase can be found in Appendix F.

where $k \in \{0, \cdots, K\}$, $K$ denotes the number of sampling steps, and $\rho$ is a constant. With $K$ and $\rho$ being properly configured, high-quality samples can be generated. The Langevin dynamics has been successfully applied to applications in computer vision [26, 52], and natural language processing [7].

We employ a single-step gradient denoising jump in this work to generate the target series. The experiments that were carried out demonstrate the effectiveness of such single-step sampling. We conduct an extra empirical study to investigate whether it is worth taking more sampling steps for further performance improvement of time series forecasting. We showcase the prediction results under different sampling strategies in Fig. 5. By omitting the additive noise in Langevin dynamics, we employ the multi-step denoising for D³VAE to generate the target series and plot the generated results in Fig. 5a. Then, with the standard Langevin dynamics, we can implement a generative procedure instead of denoising and compare the generated target series with different $\rho$ (see Figs. 5b to 5d). We can observe that more sampling steps might not be helpful in improving prediction performance for generative time series forecasting (Fig. 5a). Besides, larger sampling steps would lead to high computational complexity. On the other hand, different configurations of Langevin dynamics (with varying $\rho$) cannot bring indispensable benefits for time series forecasting (Figs. 5b to 5d).

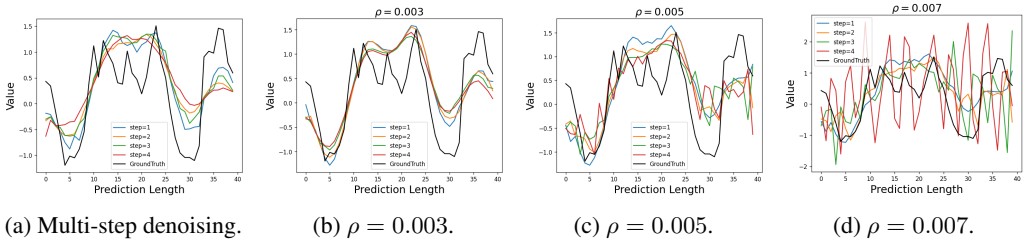

(a) Multi-step denoising.     (b) $\rho = 0.003$.     (c) $\rho = 0.005$.     (d) $\rho = 0.007$.

Figure 5: The prediction showcases in the Electricity dataset with different sampling strategies.

**Limitations.**
With the coupled diffusion probabilistic model, although the aleatoric uncertainty of the time series can be reduced, a new bias is brought into the series to mimic the distribution of the input and target. However, as a common issue in VAEs that any introduced bias in the input will result in bias in the generated output [48], the diffusion steps and variance schedule need to be chosen cautiously, such that this model can be applied to different time series tasks smoothly. The proposed model is devised for general time series forecasting, it should be used properly to avoid the potential negative societal impacts, such as illegal applications.

In time series predictive analysis, disentanglement of the latent variables has been very important for interpreting the prediction to provide more reliance. Due to the lack of prior knowledge of the entangled factors in generative time series forecasting, only unsupervised disentanglement learning can be done, which has been proven theoretically feasible for time series [31]. Despite this, for boarder applications of disentanglement and better performance, it is still worth exploring how to label the factors of time series in the future. Moreover, because of the uniqueness of time series data, it is also a promising direction to explore more generative and sampling methods for the time series generation task.

## 5    Conclusion

In this work, we propose a generative model with the bidirectional VAE as the backbone. To further improve the generalizability, we devise a coupled diffusion probabilistic model for time series forecasting. Then a scaled denoising network is developed to guarantee the prediction accuracy. Afterward, the latent variables are further disentangled for better model interpretability. Extensive experiments on synthetic data and real-world data validate that our proposed generative model achieves SOTA performance compared to existing competitive generative models.

## Acknowledgement

We thank Longyuan Power Group Corp. Ltd. for supporting this work.

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
