# A Related Work

## A.1 Time Series Forecasting

We first briefly review the related literature of time series forecasting (TSF) methods as below. Complex temporal patterns can be manifested over short- and long-term as the time series evolves across time. To leverage the time evolution nature, existing statistical models, such as ARIMA [6] and Gaussian process regression [7] have been well established and applied to many downstream tasks [28, 29, 2]. Recurrent neural network (RNN) models are also introduced to model temporal dependencies for TSF in a sequence-to-sequence paradigm [24, 9, 61, 40, 46, 50, 53]. Besides, temporal attention [49, 59, 56] and causal convolution [3, 5, 54] are further explored to model the intrinsic temporal dependencies. Recent Transformer-based models have strengthened the capability of exploring hidden intricate temporal patterns for long-term TSF [67, 42, 65, 71]. On the other hand, the multivariate nature of TSF is another topic many works have been focusing on. These works treat a collection of time series as a unified entity and mine the inter-series correlations with different techniques, such as probabilistic models [53, 50], matrix/tensor factorization [54, 55], convolution neural networks (CNNs) [3, 40], and graph neural networks (GNNs) [26, 43, 70, 66, 8].

To improve the reliability and performance of TSF, instead of modeling the raw data, there exist works inferring the underlying distribution of the time series data with generative models [69, 14]. Many studies have employed a variational auto-encoder (VAE) to model the probabilistic distribution of sequential data [21, 23, 12, 10, 47]. For example, VRNN [12] employs the VAE to each hidden state of RNN such that the variability of highly structured sequential data can be captured. To yield predictive distribution for multivariate TSF, TLAE [47] implements nonlinear transformation by replacing matrix factorization with encoder-decoder architecture and temporal deep temporal latent model. Another line of generative methods for TSF focus on energy-based models (EBMs), such as TimeGrad [51] and ScoreGrad [68]. EBMs do not restrict the tractability of the normalizing constants [68]. Though flexible, the unknown normalizing constant makes the training of EBMs particularly difficult.

This paper focuses on TSF with VAE-based models. Besides, as many real-world time series data are relatively short and small [58], a coupled probabilistic diffusion model is proposed to augment the input series, as well as the output series, simultaneously, such that the distribution space can be enlarged without increasing the aleatoric uncertainty [34]. Moreover, to guarantee the generated target series moving toward the true target, a multi-scaled score-matching denoising network is plugged in for accurate future series prediction. To our knowledge, this is the first work focusing on generative TSF with the diffusion model and denoising techniques.

## A.2 Time Series Augmentation

Both the traditional methods and deep learning methods can deteriorate when limited time series data are encountered. Generating synthetic time series is commonly adopted for augmenting short time series [13, 18, 69]. Transforming the original time series by cropping, flipping, and warping [32, 15] is another approach dedicated to TSF when the training data is limited. Whereas the synthetic time series may not respect the original feature relationship across time, and the transformation methods do not change the distribution space. Thus, the overfitting issues cannot be avoided. Incorporating the probabilistic diffusion model for TSF differentiates our work from existing time series augmentation methods.

## A.3 Uncertainty Estimation and Denoising for Time Series Forecasting

There exist works aiming to estimate the uncertainty [34] for time series forecasting [48, 62, 25] by epistemic uncertainty. Nevertheless, the inevitable aleatoric uncertainty of time series is often ignored, which may stem from error-prone data measurement, collection, and so forth [63]. Another line of studies focuses on detecting noise in time series data [45] or devising suitable models for noise alleviation [22]. However, none of the existing works attempts to quantify the aleatoric uncertainty, which further differentiates our work from priors.

It is necessary to relieve the effect of noise in real-world time series data [16]. [4, 39] propose to preprocess the time series with smoothing and filtering techniques. However, such preprocessing methods can only be applied to the noise raised by the irregular data of time series. Neural networks

are also introduced to denoise the time series [20, 57, 22, 33], while these deep networks can only deal with specific types of time series as well.

### A.4 Interpretability of Time Series Forecasting

A number of works put effort into explaining the deep neural networks [64, 35, 1] to make the prediction more interpretable, but these methods often lack reliability when the explanation is sensitive to factors that do not contribute to the prediction [37]. Several works have been proposed to increase the reliability of TSF tasks [30, 31]. For multivariate time series, the interpretability of the representations can be improved by mapping the time series into latent space [19]. Besides, recent works have been proposed to disentangle the latent variables to identify the independent factors of the data, which can further lead to improved interpretability of the representation and higher performance [27, 41, 36]. The disentangled VAE has been applied to time series to benefit the generated results [44]. However, the choice of the latent variables is crucial for the disentanglement of time series data. We devise a bidirectional VAE (BVAE) and take the dimensions of each latent variable as the factors to be disentangled.

## B  Proofs of Lemma 1 and Lemma 2

With the coupled diffusion process and Eqs. (5) and (6), as well as Proposition 1, introduced in the main text, the diffused target series and generated target series can be decomposed as $\widetilde{Y}^{(t)} = \langle \widetilde{Y}_r^{(t)}, \delta_{\widetilde{Y}}^{(t)} \rangle$ and $\widehat{Y}^{(t)} = \langle \widehat{Y}^{(t)}, \delta_{\widehat{Y}}^{(t)} \rangle$. Then, we can draw the following two conclusions:

**Lemma 1.** $\forall \varepsilon > 0$, there exists a probabilistic model $f_{\phi,\theta} := (p_\phi, p_\theta)$ to guarantee that $\mathcal{D}_{\mathrm{KL}}(q(\widetilde{Y}_r^{(t)})||p_\theta(\widehat{Y}_r^{(t)})) < \varepsilon$, where $\widehat{Y}_r^{(t)} = f_{\phi,\theta}(X^{(t)})$.

*Proof.* According to Proposition 1, $\widehat{Y}_r$ can be fully captured by the model. That is, $\|Y_r - \widehat{Y}_r\| \longrightarrow 0$ where $Y_r$ is the ideal part of ground truth target series $Y$. And, with Eq. (6) (in the main text), $\widetilde{Y}_r^{(t)} = \sqrt{\bar{\alpha}_t'} Y_r$. Therefore, $\|\widetilde{Y}_r^{(t)} - \widehat{Y}_r^{(t)}\| \longrightarrow 0$ when $t \to \infty$. $\square$

**Lemma 2.** *With the coupled diffusion process, the difference between diffusion noise and generation noise will be reduced, i.e.,* $\lim_{t\to\infty} \mathcal{D}_{\mathrm{KL}}(q(\delta_{\widetilde{Y}}^{(t)})||p_\theta(\delta_{\widehat{Y}}^{(t)}|Z^{(t)})) < \mathcal{D}_{\mathrm{KL}}(q(\epsilon_Y)||p_\theta(\epsilon_{\widehat{Y}}))$.

*Proof.* According to Proposition 1, the noise of $Y$ consists of the estimation noise $\epsilon_{\widehat{Y}}$ and residual noise $\delta_Y$, i.e., $\epsilon_Y = \langle \epsilon_{\widehat{Y}}, \delta_Y \rangle$ where $\epsilon_{\widehat{Y}}$ and $\delta_Y$ are independent of each other, then $q(\epsilon_Y) = q(\epsilon_{\widehat{Y}})q(\delta_Y)$. Let $\Delta = \mathcal{D}_{\mathrm{KL}}(q(\epsilon_Y)||p_\theta(\epsilon_{\widehat{Y}})) - \mathcal{D}_{\mathrm{KL}}(q(\epsilon_{\widehat{Y}})||p_\theta(\epsilon_{\widehat{Y}}))$, we have

$$\begin{aligned} \Delta &= \mathcal{D}_{\mathrm{KL}}(q(\epsilon_{\widehat{Y}})q(\delta_Y)||p_\theta(\epsilon_{\widehat{Y}})) - \mathcal{D}_{\mathrm{KL}}(q(\epsilon_{\widehat{Y}})||p_\theta(\epsilon_{\widehat{Y}})) \\ &= \mathcal{D}_{\mathrm{KL}}(q(\epsilon_{\widehat{Y}})||p_\theta(\epsilon_{\widehat{Y}})) + \mathcal{D}_{\mathrm{KL}}(q(\delta_Y)||p_\theta(\epsilon_{\widehat{Y}})) - \mathcal{D}_{\mathrm{KL}}(q(\epsilon_{\widehat{Y}})||p_\theta(\epsilon_{\widehat{Y}})) \\ &= \mathcal{D}_{\mathrm{KL}}(q(\delta_Y)||p_\theta(\epsilon_{\widehat{Y}})) > 0, \end{aligned}$$

which leads to $\mathcal{D}_{\mathrm{KL}}(q(\epsilon_Y)||p_\theta(\epsilon_{\widehat{Y}})) > \mathcal{D}_{\mathrm{KL}}(q(\epsilon_{\widehat{Y}})||p_\theta(\epsilon_{\widehat{Y}}) > 0$. Moreover, both $\delta_{\widetilde{Y}}^{(t)}$ and $\delta_{\widehat{Y}}^{(t)}$ are Gaussian noises, when $t \to \infty$, $\exists \varepsilon' > 0$, we have $\mathcal{D}_{\mathrm{KL}}(q(\delta_{\widetilde{Y}}^{(t)})||p_\theta(\delta_{\widehat{Y}}^{(t)}|Z^{(t)})) \leq \varepsilon' < \mathcal{D}_{\mathrm{KL}}(q(\epsilon_Y)||p_\theta(\epsilon_{\widehat{Y}}))$. $\square$

## C  Extra Implementation Details

### C.1  Experimental Settings

**Datasets Description.** The main descriptive statistics of the real-world datasets adopted in the experiments of this work are demonstrated in Table 5.

**Input Representation.** We adopt the embedding method introduced in [71] and feed it to an RNN to extract the temporal dependency. Then we concatenate them as follows:

$$X_{\text{input}} = \text{CONCAT}(\text{RNN}(\mathcal{E}(X)), \mathcal{E}(X)),$$

Table 5: Statistical descriptions of the real-world datasets.

| Datasets | # Dims. | Full Data | | Sliced Data | | Target Variable | Time Interval |
|---|---|---|---|---|---|---|---|
| | | Time Span | # Points | Pct. of Full Data | # Points | | |
| Traffic | 862 | 2015-2016 | 17544 | 5% | 877 | Sensor 862 | 1 hour |
| Electricity | 321 | 2011-2014 | 18381 | 3% | 551 | MT_321 | 10 mins |
| Weather | 21 | 2020-2021 | 36761 | 2% | 735 | CO2 (ppm) | 10 mins |
| ETTm1 | 7 | 2016-2018 | 69680 | 1% | 697 | OT | 15 mins |
| ETTh1 | 7 | 2016-2018 | 17420 | 5% | 871 | OT | 1 hour |
| Wind | 7 | 2020-2021 | 45550 | 2% | 911 | wind_power | 15 mins |

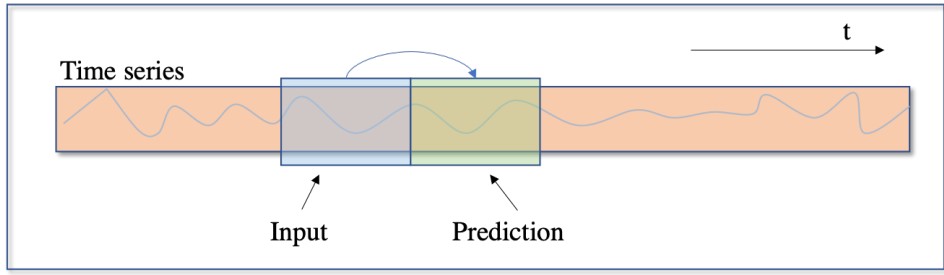

Figure 6: Forecasting process of DeepAR, TimeGrad, and GP-copula. The sliding step is set to 1.

where $X$ is the raw time series data and $\mathcal{E}(\cdot)$ denotes the embedding operation. Here, we use a two-layer gated recurrent unit (GRU), and the dimensionality of the hidden state and embeddings are $128$ and $64$, respectively.

**Diffusion Process Configuration.** Besides, the diffusion process is configured to be $\beta_t \in [0, 0.1]$ and $T = 100$ for the **Weather** dataset, $\beta_t \in [0, 0.1]$ and $T = 1000$ for the **ETTh1** dataset, $\beta_t \in [0, 0.08]$ and $T = 1000$ for the **Wind** dataset, and $\beta_t \in [0, 0.01]$ and $T = 1000$ for the other datasets.

### C.2 Implementation Details of Baselines

We select previous state-of-the-art generative models as our baselines in the experiments on synthetic and real-world datasets. Specifically, **1) GP-copula** [52] is a method based on the Gaussian process, which is devoted to high-dimensional multivariate time series, **2) DeepAR** [53] combines traditional auto-regressive models with RNNs by modeling a probabilistic distribution in an auto-encoder fashion, **3) TimeGrad** [51] is an auto-regressive model for multivariate probabilistic time series forecasting with the help of an energy-based model, **4) Vanilla VAE** (VAE for short) [38] is a classical statistical variational inference method on top of auto-encoder, **5) NVAE** [60] is a deep hierarchical VAE built for image generation using depth-wise separable convolutions and batch normalization, **6) factor-VAE** (f-VAE for short) [36] disentangles the latent variables by encouraging the distribution of representations to be factorial and independent across dimensions, and **7)** $\beta$**-TCVAE** [11] learns the disentangled representations with total correlation variational auto-encoder algorithm.

To train DeepAR, TimeGrad, and GP-copula in accordance with their original settings, the batch is constructed without shuffling the samples. The instances (sampled with the input-$l_x$-predict-$l_y$ rolling window and $l_x = l_y$, as illustrated in Fig. 6) are fed to the training procedure of these three baselines in chronological order. Besides, these three baselines employ the cumulative distribution function (CDF) for training, so the CDF needs to be reverted to the real distribution for testing.

For f-VAE, $\beta$-TCVAE, and VAE, since the dimensionality of different time series varies, we design a preprocess block to map the original time series into a tensor with the fix-sized dimensionality, which can further suit the VAEs well. The preprocess block consists of three nonlinear layers with the sizes of the hidden states: $\{128, 64, 32\}$. For NVAE, we keep the original settings suggested in [60] and use Gaussian distribution as the prior. All the baselines are trained using early stopping, and the patience is set to 5.

Table 6: Performance comparisons of short-term and long-term TSF in real-world datasets in terms of MSE and CRPS. For MSE and CRPS, the lower, the better. The best results are in boldface.

| Model | | $D^3$VAE | NVAE | $\beta$-TCVAE | f-VAE | DeepAR | TimeGrad | GP-copula | VAE |
|---|---|---|---|---|---|---|---|---|---|
| **Traffic** | 8 | $\mathbf{0.081}_{\pm.003}$ | $1.300_{\pm.024}$ | $1.003_{\pm.006}$ | $0.982_{\pm.059}$ | $3.895_{\pm.306}$ | $3.695_{\pm.246}$ | $4.299_{\pm.372}$ | $0.794_{\pm.130}$ |
| | | $\mathbf{0.207}_{\pm.003}$ | $0.593_{\pm.004}$ | $0.894_{\pm.003}$ | $0.666_{\pm.032}$ | $1.391_{\pm.071}$ | $1.410_{\pm.027}$ | $1.408_{\pm.046}$ | $0.759_{\pm.07}$ |
| | 16 | $\mathbf{0.081}_{\pm.009}$ | $1.271_{\pm.019}$ | $0.997_{\pm.004}$ | $0.998_{\pm.042}$ | $4.140_{\pm.320}$ | $3.495_{\pm.362}$ | $4.575_{\pm.141}$ | $0.632_{\pm.057}$ |
| | | $\mathbf{0.200}_{\pm.014}$ | $0.589_{\pm.001}$ | $0.893_{\pm.002}$ | $0.692_{\pm.026}$ | $1.338_{\pm.043}$ | $1.329_{\pm.057}$ | $1.506_{\pm.025}$ | $0.671_{\pm.038}$ |
| | 32 | $\mathbf{0.091}_{\pm.007}$ | $0.126_{\pm.013}$ | $1.254_{\pm0.019}$ | $0.977_{\pm.002}$ | $4.234_{\pm.139}$ | $5.195_{\pm2.26}$ | $3.717_{\pm.361}$ | $0.735_{\pm.084}$ |
| | | $\mathbf{0.216}_{\pm.012}$ | $0.422_{\pm.012}$ | $0.937_{0.007}$ | $0.882_{\pm.001}$ | $1.367_{\pm.015}$ | $1.565_{\pm.329}$ | $1.342_{\pm.048}$ | $0.735_{\pm.048}$ |
| | 64 | $\mathbf{0.125}_{\pm.005}$ | $1.263_{\pm0.014}$ | $0.903_{\pm.111}$ | $0.936_{\pm.190}$ | $3.381_{\pm.130}$ | $3.692_{\pm1.54}$ | $3.492_{\pm.092}$ | $0.692_{\pm.059}$ |
| | | $\mathbf{0.244}_{\pm.006}$ | $0.940_{\pm0.005}$ | $0.839_{\pm.062}$ | $0.829_{\pm.078}$ | $1.233_{\pm.027}$ | $1.412_{\pm0.257}$ | $1.367_{\pm.012}$ | $0.710_{\pm.035}$ |
| **Electricity** | 8 | $\mathbf{0.251}_{\pm.015}$ | $1.134_{\pm.029}$ | $0.901_{\pm.052}$ | $0.893_{\pm.069}$ | $2.934_{\pm.173}$ | $2.703_{\pm.087}$ | $2.924_{\pm.218}$ | $0.853_{\pm.040}$ |
| | | $\mathbf{0.398}_{\pm.011}$ | $0.542_{\pm.003}$ | $0.831_{\pm.004}$ | $0.809_{\pm.024}$ | $1.244_{\pm.037}$ | $1.208_{\pm.024}$ | $1.249_{\pm.048}$ | $0.795_{\pm.016}$ |
| | 16 | $\mathbf{0.308}_{\pm.030}$ | $1.150_{\pm.032}$ | $0.850_{\pm.003}$ | $0.807_{\pm.034}$ | $2.803_{\pm.199}$ | $2.770_{\pm.237}$ | $3.065_{\pm.186}$ | $0.846_{\pm.062}$ |
| | | $\mathbf{0.437}_{\pm.020}$ | $0.531_{\pm.003}$ | $0.814_{\pm.002}$ | $0.782_{\pm.024}$ | $1.220_{\pm.048}$ | $1.240_{\pm.048}$ | $1.307_{\pm.042}$ | $0.793_{\pm.029}$ |
| | 32 | $\mathbf{0.410}_{\pm.075}$ | $1.302_{\pm0.011}$ | $0.844_{\pm.025}$ | $0.861_{\pm.105}$ | $2.402_{\pm.156}$ | $2.640_{\pm.138}$ | $2.880_{\pm.221}$ | $0.841_{\pm.071}$ |
| | | $\mathbf{0.534}_{\pm.058}$ | $0.944_{\pm0.005}$ | $0.808_{\pm.005}$ | $0.797_{\pm.037}$ | $1.130_{\pm.055}$ | $1.234_{\pm.027}$ | $1.281_{\pm.054}$ | $0.790_{\pm.026}$ |
| **Weather** | 8 | $\mathbf{0.169}_{\pm.022}$ | $0.801_{\pm.024}$ | $0.234_{\pm.042}$ | $0.591_{\pm.198}$ | $2.317_{\pm.357}$ | $2.715_{\pm.189}$ | $2.412_{\pm.761}$ | $0.560_{\pm.192}$ |
| | | $\mathbf{0.357}_{\pm.024}$ | $0.757_{\pm.013}$ | $0.404_{\pm.040}$ | $0.565_{\pm.080}$ | $0.858_{\pm.078}$ | $0.920_{\pm.013}$ | $0.897_{\pm.115}$ | $0.572_{\pm.077}$ |
| | 16 | $\mathbf{0.187}_{\pm.047}$ | $0.811_{\pm.016}$ | $0.212_{\pm.012}$ | $0.530_{\pm.167}$ | $1.269_{\pm.187}$ | $1.110_{\pm.083}$ | $1.357_{\pm.145}$ | $0.424_{\pm.141}$ |
| | | $\mathbf{0.361}_{\pm.046}$ | $0.759_{\pm.009}$ | $0.388_{\pm.014}$ | $0.547_{\pm.067}$ | $0.783_{\pm.059}$ | $0.733_{\pm.016}$ | $0.811_{\pm.032}$ | $0.503_{\pm.068}$ |
| | 32 | $\mathbf{0.203}_{\pm.008}$ | $0.836_{\pm0.014}$ | $0.439_{\pm.394}$ | $0.337_{\pm.086}$ | $2.518_{\pm.546}$ | $1.178_{\pm.069}$ | $1.065_{\pm.145}$ | $0.329_{\pm.083}$ |
| | | $\mathbf{0.383}_{\pm.007}$ | $0.777_{\pm0.007}$ | $0.508_{\pm.176}$ | $0.461_{\pm.031}$ | $0.847_{\pm.036}$ | $0.724_{\pm.021}$ | $0.747_{\pm.035}$ | $0.459_{\pm.045}$ |
| | 64 | $\mathbf{0.191}_{\pm.022}$ | $0.932_{0.020}$ | $0.276_{\pm.026}$ | $0.676_{\pm.484}$ | $3.595_{\pm.956}$ | $1.063_{\pm.061}$ | $0.992_{\pm.114}$ | $0.721_{\pm.496}$ |
| | | $\mathbf{0.358}_{\pm.044}$ | $0.836_{0.009}$ | $0.463_{\pm.026}$ | $0.612_{\pm.176}$ | $0.994_{\pm.100}$ | $0.696_{\pm.011}$ | $0.699_{\pm.016}$ | $0.635_{\pm.204}$ |
| **ETTm1** | 8 | $\mathbf{0.527}_{\pm.073}$ | $0.921_{\pm.026}$ | $1.538_{\pm.254}$ | $2.326_{\pm.445}$ | $2.204_{\pm.420}$ | $1.877_{\pm.245}$ | $2.024_{\pm.143}$ | $2.375_{\pm.405}$ |
| | | $\mathbf{0.557}_{0.048}$ | $0.760_{\pm.026}$ | $1.015_{\pm.112}$ | $1.260_{\pm.167}$ | $0.984_{\pm.074}$ | $0.908_{\pm.038}$ | $0.961_{\pm.027}$ | $1.258_{\pm.104}$ |
| | 16 | $\mathbf{0.968}_{\pm.104}$ | $1.100_{\pm.032}$ | $1.744_{\pm.100}$ | $2.339_{\pm.270}$ | $2.350_{\pm.170}$ | $2.032_{\pm.234}$ | $2.486_{\pm.207}$ | $2.321_{\pm.469}$ |
| | | $\mathbf{0.821}_{\pm.072}$ | $0.822_{\pm.026}$ | $1.104_{\pm.041}$ | $1.249_{\pm.088}$ | $0.974_{\pm.016}$ | $0.919_{\pm.031}$ | $0.984_{\pm.016}$ | $1.259_{\pm.132}$ |
| | 32 | $\mathbf{0.707}_{\pm.061}$ | $1.298_{\pm.028}$ | $1.438_{\pm.429}$ | $2.563_{\pm.358}$ | $4.855_{\pm.179}$ | $1.251_{\pm.133}$ | $1.402_{\pm.187}$ | $2.660_{\pm.349}$ |
| | | $\mathbf{0.697}_{\pm.040}$ | $0.893_{\pm.010}$ | $0.953_{\pm.173}$ | $1.330_{\pm.104}$ | $1.787_{\pm.029}$ | $0.822_{\pm.032}$ | $0.844_{\pm.043}$ | $1.367_{\pm.083}$ |
| **ETTh1** | 8 | $\mathbf{0.292}_{\pm.036}$ | $0.483_{\pm.017}$ | $0.703_{\pm.054}$ | $0.870_{\pm.134}$ | $3.451_{\pm.335}$ | $4.259_{\pm1.13}$ | $4.278_{\pm1.12}$ | $1.006_{\pm.281}$ |
| | | $\mathbf{0.424}_{\pm.033}$ | $0.461_{\pm.011}$ | $0.644_{\pm.038}$ | $0.730_{\pm.060}$ | $1.194_{\pm.034}$ | $1.092_{\pm.028}$ | $1.169_{\pm.055}$ | $0.762_{\pm.115}$ |
| | 16 | $\mathbf{0.374}_{\pm.061}$ | $0.488_{\pm.010}$ | $0.681_{\pm.018}$ | $0.983_{\pm.139}$ | $1.929_{\pm.105}$ | $1.332_{\pm.125}$ | $1.701_{\pm.088}$ | $0.681_{\pm.104}$ |
| | | $0.488_{\pm.039}$ | $\mathbf{0.463}_{\pm.018}$ | $0.640_{\pm.008}$ | $0.760_{\pm.062}$ | $1.029_{\pm.030}$ | $0.879_{\pm.037}$ | $0.999_{\pm.023}$ | $0.641_{\pm.055}$ |
| | 32 | $\mathbf{0.334}_{\pm.008}$ | $0.464_{\pm0.007}$ | $0.477_{\pm.035}$ | $0.669_{\pm.092}$ | $6.153_{\pm.715}$ | $1.514_{\pm.042}$ | $1.922_{\pm.032}$ | $0.578_{\pm.062}$ |
| | | $\mathbf{0.461}_{\pm.004}$ | $0.543_{\pm0.004}$ | $0.537_{\pm.019}$ | $0.646_{\pm.048}$ | $1.689_{\pm.112}$ | $0.925_{\pm.016}$ | $1.068_{\pm.011}$ | $0.597_{\pm.035}$ |
| | 64 | $0.349_{\pm.039}$ | $0.425_{\pm.006}$ | $0.418_{\pm.021}$ | $0.484_{\pm.051}$ | $2.419_{\pm.520}$ | $1.150_{0.118}$ | $1.654_{\pm.117}$ | $0.463_{\pm.081}$ |
| | | $0.473_{\pm.024}$ | $0.523_{0.004}$ | $0.517_{\pm.013}$ | $0.551_{\pm.027}$ | $1.223_{\pm.127}$ | $0.835_{\pm.045}$ | $0.987_{\pm.036}$ | $0.546_{\pm.042}$ |
| **Wind** | 8 | $\mathbf{0.681}_{\pm.075}$ | $1.854_{\pm.032}$ | $1.321_{\pm.379}$ | $1.942_{\pm.101}$ | $12.53_{\pm2.25}$ | $12.67_{\pm1.75}$ | $11.35_{\pm6.61}$ | $2.006_{\pm.145}$ |
| | | $\mathbf{0.596}_{\pm.052}$ | $1.223_{\pm.014}$ | $0.863_{\pm.143}$ | $1.067_{\pm.086}$ | $1.370_{\pm.107}$ | $1.440_{\pm.059}$ | $1.305_{\pm.369}$ | $1.103_{\pm.100}$ |
| | 16 | $1.033_{\pm.062}$ | $1.955_{\pm.015}$ | $\mathbf{0.894}_{\pm.038}$ | $1.262_{\pm.178}$ | $13.96_{\pm1.53}$ | $12.86_{\pm2.60}$ | $13.79_{\pm5.37}$ | $1.138_{\pm.205}$ |
| | | $0.757_{\pm.053}$ | $1.247_{\pm.011}$ | $0.785_{\pm.037}$ | $0.843_{\pm.066}$ | $1.347_{\pm.060}$ | $1.240_{\pm.070}$ | $1.261_{\pm.171}$ | $0.862_{\pm.092}$ |
| | 32 | $\mathbf{1.224}_{\pm.060}$ | $1.784_{\pm.011}$ | $1.266_{\pm.006}$ | $1.434_{\pm.126}$ | $5.398_{\pm.179}$ | $13.10_{\pm.955}$ | $15.33_{1.904}$ | $1.480_{\pm.072}$ |
| | | $\mathbf{0.869}_{\pm.074}$ | $1.200_{\pm.007}$ | $0.872_{\pm.010}$ | $0.920_{\pm.077}$ | $1.434_{\pm.013}$ | $1.518_{\pm.020}$ | $1.614_{\pm.118}$ | $0.987_{\pm.010}$ |
| | 64 | $0.902_{\pm.024}$ | $1.652_{\pm.010}$ | $\mathbf{0.786}_{\pm.022}$ | $0.898_{\pm.095}$ | $4.403_{\pm.301}$ | $3.857_{\pm.597}$ | $3.564_{\pm.293}$ | $1.374_{\pm1.02}$ |
| | | $0.761_{\pm.021}$ | $1.167_{\pm.005}$ | $\mathbf{0.742}_{\pm.017}$ | $0.789_{\pm.048}$ | $1.361_{\pm.021}$ | $1.110_{\pm.143}$ | $1.152_{\pm.081}$ | $0.842_{\pm.215}$ |

Table 7: Performance comparisons of TSF in 100%-Electricity and 100%-ETTm1 datasets in terms of MSE and CRPS. The best results are highlighted in boldface.

| Model | | $D^3$VAE | NVAE | $\beta$-TCVAE | f-VAE | DeepAR | TimeGrad | GP-copula | VAE |
|---|---|---|---|---|---|---|---|---|---|
| Electricity | 16 | $\mathbf{0.330}_{\pm.033}$ | $1.408_{\pm.015}$ | $0.801_{\pm.001}$ | $0.765_{\pm.026}$ | $33.93_{\pm1.85}$ | $46.69_{\pm3.13}$ | $50.25_{\pm4.39}$ | $0.680_{\pm.022}$ |
| | | $\mathbf{0.445}_{\pm.020}$ | $0.999_{\pm.006}$ | $0.723_{\pm.001}$ | $0.710_{\pm.013}$ | $2.650_{\pm.030}$ | $2.702_{\pm.079}$ | $2.796_{\pm.072}$ | $0.675_{\pm.008}$ |
| | 32 | $\mathbf{0.336}_{\pm.017}$ | $1.403_{\pm.014}$ | $0.802_{\pm.001}$ | $0.748_{\pm.033}$ | $46.10_{\pm2.00}$ | $30.94_{\pm1.70}$ | $32.13_{\pm1.96}$ | $0.727_{\pm.033}$ |
| | | $\mathbf{0.444}_{\pm.015}$ | $0.997_{\pm.007}$ | $0.724_{\pm.001}$ | $0.703_{\pm.016}$ | $2.741_{\pm.011}$ | $2.476_{\pm.042}$ | $2.591_{\pm.064}$ | $0.692_{\pm.014}$ |
| ETTm1 | 16 | $\mathbf{0.018}_{\pm.002}$ | $2.577_{\pm.047}$ | $0.918_{\pm.015}$ | $1.285_{\pm.236}$ | $73.82_{\pm3.25}$ | $68.26_{\pm2.04}$ | $66.97_{\pm2.02}$ | $1.335_{\pm.156}$ |
| | | $\mathbf{0.102}_{\pm.003}$ | $1.509_{0.016}$ | $0.766_{\pm.005}$ | $0.911_{\pm.090}$ | $1.136_{\pm.013}$ | $1.153_{\pm.019}$ | $1.111_{\pm.016}$ | $0.923_{\pm.056}$ |
| | 32 | $\mathbf{0.034}_{\pm.001}$ | $2.622_{\pm.057}$ | $0.929_{\pm.010}$ | $1.420_{\pm.073}$ | $68.11_{\pm2.60}$ | $53.47_{\pm26.1}$ | $63.67_{\pm1.14}$ | $1.223_{\pm.213}$ |
| | | $\mathbf{0.144}_{\pm.006}$ | $1.524_{\pm.018}$ | $0.770_{\pm.004}$ | $0.960_{\pm.021}$ | $1.121_{\pm.024}$ | $1.083_{\pm.109}$ | $1.097_{\pm.008}$ | $0.888_{\pm.082}$ |

# D    Supplementary Experimental Results

## D.1    Comparisons of Predictive Performance for TSF Under Different Settings

**Longer-Term Time Series Forecasting.** To further inspect the performance of our method, we additionally conduct more experiments for longer-term time series forecasting. In particular, by configuring the output length with 32 and 64[1], we compare $D^3$VAE to other baselines in terms of MSE and CRPS, and the results (including short-term and long-term) are reported in Table 6. We can conclude that $D^3$VAE also outperforms the competitive baselines consistently under the longer-term forecasting settings.

**Time Series Forecasting in Full Datasets.** Moreover, we evaluate the predictive performance for time series forecasting in two "full-version" datasets, i.e. 100%-Electricity and 100%-ETTm1. The split of train/validation/test is 7:1:2 which is the same as the main experiments. The comparisons in terms of MSE and CRPS can be found in Table 7. With sufficient data, compared to previous state-of-the-art generative models, the MSE and CRPS reductions of our method are also satisfactory under different settings (including input-16-predict-16 and input-32-predict-32). For example, in the Electricity dataset, compared to the second best results, $D^3$VAE achieves 52% ($0.680 \rightarrow 0.330$) and 54% ($0.727 \rightarrow 0.336$) MSE reductions, and 34% ($0.675 \rightarrow 0.445$) and 36% ($0.692 \rightarrow 0.444$) CRPS reductions, under input-16-predict-16 and input-32-predict-32 settings, respectively.

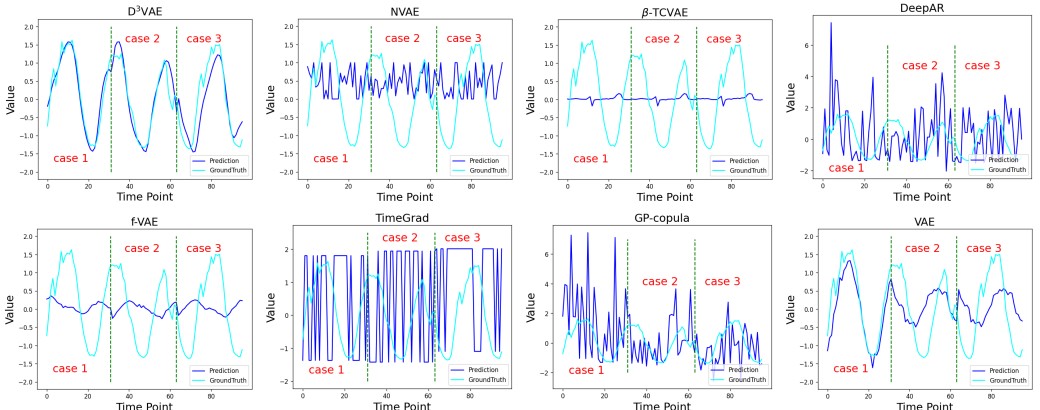

Figure 7: The case study of forecasting results on the Traffic dataset under input-32-predict-32 settings. Only the last dimension is plotted. To demonstrate the forecasting results in a long range, we show the predictions of three cases ordered chronologically without overlapping.

---
[1]The length of the input time series is the same as the output time series.

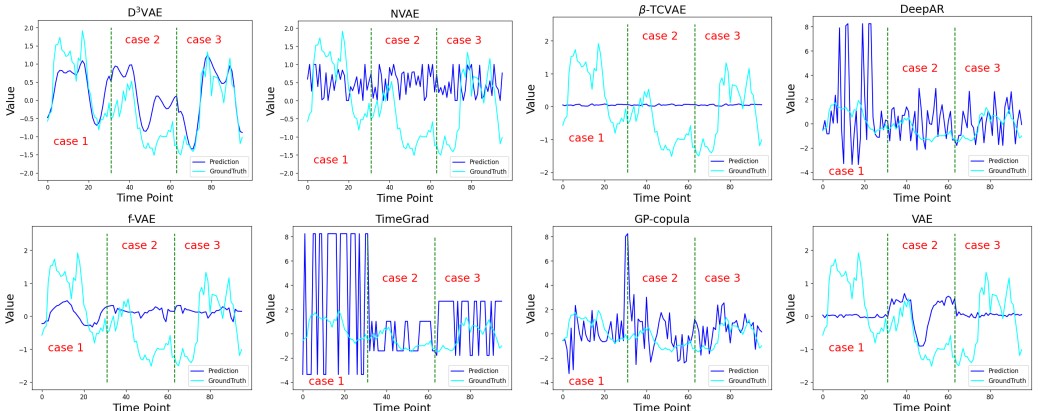

Figure 8: Case study of the forecasting results from the Electricity dataset (same settings as Fig. 7).

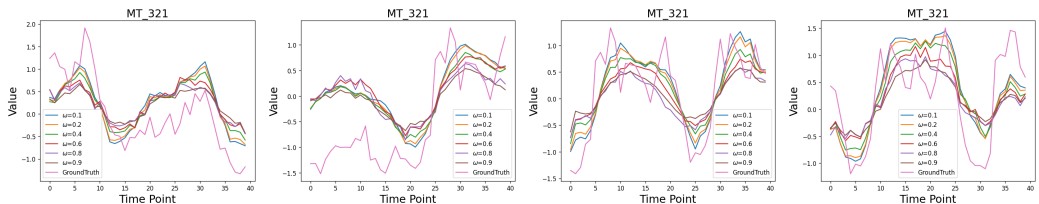

Figure 9: Forecasting results (under the input-40-predict-40 setting) of a case from the Electricity dataset with $\omega$ increasing from 0.1 to 0.9.

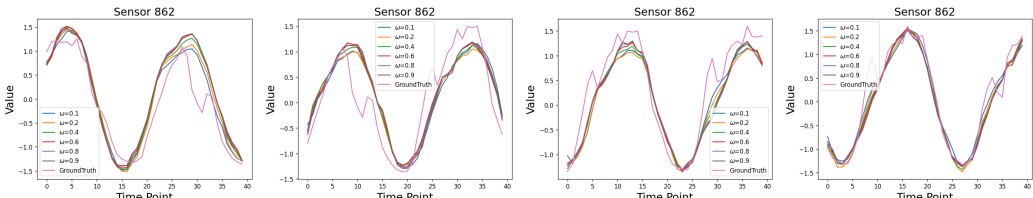

Figure 10: Forecasting results of a case from the Traffic dataset under the input-40-predict-40 setting.

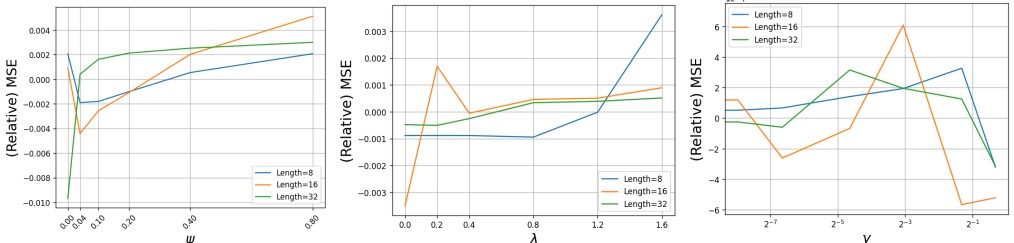

Figure 11: Sensitivity analysis of the trade-off hyperparameters in reconstruction loss $\mathcal{L}$. To highlight the changes in prediction performance against hyperparameters, the relative value of MSE is used.

## D.2 Case Study

We showcase the prediction results of our model and seven baseline models on the Traffic and Electricity datasets in Figs. 7 and 8. Our model can provide the most accurate forecasting results regarding trends and variations.

## D.3 The Effect of Scale Parameter $\omega$

We demonstrate the forecasting results with different values of $\omega$ on Electricity and Traffic datasets, and the results are plotted in Figs. 9 and 10. It can be depicted that larger or smaller $\omega$ would lead

Table 8: Performance comparisons of D$^3$VAE w.r.t. varying the length of (input and output) time series and the data size. The results are reported on the Electricity dataset.

| Length | Metric | Percentage of Full Electricity Data | | | | | | |
|---|---|---|---|---|---|---|---|---|
| | | 100% | 80% | 60% | 40% | 20% | 10% | 5% |
| 8 | MSE | $0.258_{\pm.019}$ | $0.227_{\pm.016}$ | $0.368_{\pm.019}$ | $0.389_{\pm.034}$ | $3.861_{\pm.480}$ | $0.693_{\pm.223}$ | $0.206_{\pm.018}$ |
| | CRPS | $0.383_{\pm.015}$ | $0.355_{\pm.015}$ | $0.453_{\pm.009}$ | $0.504_{\pm.034}$ | $1.728_{\pm.110}$ | $0.673_{\pm.132}$ | $0.352_{\pm.016}$ |
| 16 | MSE | $0.330_{\pm.033}$ | $0.253_{\pm.018}$ | $0.343_{\pm.024}$ | $0.463_{\pm.089}$ | $4.428_{\pm.694}$ | $0.401_{\pm.068}$ | $0.247_{\pm.056}$ |
| | CRPS | $0.445_{\pm.020}$ | $0.373_{\pm.014}$ | $0.433_{\pm.015}$ | $0.562_{\pm.049}$ | $1.858_{\pm.147}$ | $0.496_{\pm.047}$ | $0.378_{\pm.036}$ |
| 32 | MSE | $0.336_{\pm.017}$ | $0.300_{\pm.039}$ | $0.484_{\pm.048}$ | $0.739_{\pm.209}$ | $5.029_{\pm.811}$ | $0.884_{\pm.237}$ | $0.304_{\pm.094}$ |
| | CRPS | $0.444_{\pm.015}$ | $0.413_{\pm.034}$ | $0.537_{\pm.025}$ | $0.693_{\pm.099}$ | $1.989_{\pm.172}$ | $0.723_{\pm.112}$ | $0.418_{\pm.065}$ |

Table 9: Performance comparisons of D$^3$VAE w.r.t. varying the length of (input and output) time series and the data size. The results are reported on the Traffic dataset.

| Length | Metric | Percentage of Full Traffic Data | | | | | | |
|---|---|---|---|---|---|---|---|---|
| | | 100% | 80% | 60% | 40% | 20% | 10% | 5% |
| 8 | MSE | $0.370_{\pm.021}$ | $0.215_{\pm.016}$ | $0.063_{\pm.002}$ | $0.062_{\pm.002}$ | $0.054_{\pm.004}$ | $0.210_{\pm.012}$ | $0.081_{\pm.003}$ |
| | CRPS | $0.415_{\pm.013}$ | $0.347_{\pm.015}$ | $0.184_{\pm.003}$ | $0.179_{\pm.005}$ | $0.172_{\pm.008}$ | $0.251_{\pm.005}$ | $0.207_{\pm.003}$ |
| 16 | MSE | $0.272_{\pm.007}$ | $0.189_{\pm.006}$ | $0.063_{\pm.001}$ | $0.058_{\pm.003}$ | $0.056_{\pm.003}$ | $0.178_{\pm.006}$ | $0.081_{\pm.009}$ |
| | CRPS | $0.334_{\pm.009}$ | $0.321_{\pm.008}$ | $0.180_{\pm.002}$ | $0.168_{\pm.006}$ | $0.169_{\pm.005}$ | $0.239_{\pm.007}$ | $0.200_{\pm.003}$ |
| 32 | MSE | $0.307_{\pm.015}$ | $0.197_{\pm.005}$ | $0.064_{\pm.002}$ | $0.063_{\pm.002}$ | $0.056_{\pm.004}$ | $0.191_{\pm.011}$ | $0.091_{\pm.007}$ |
| | CRPS | $0.363_{\pm.008}$ | $0.335_{\pm.004}$ | $0.179_{\pm.002}$ | $0.179_{\pm.003}$ | $0.170_{\pm.005}$ | $0.235_{\pm.008}$ | $0.216_{\pm.012}$ |

to deviated prediction, which is far from the ground truth. Therefore, the value of $\omega$ does affect the prediction performance, which should be tuned properly.

### D.4 Sensitivity Analysis of Trade-off Parameters in Reconstruction Loss $\mathcal{L}$

To examine the effect of the trade-off hyperparameters in loss $\mathcal{L}$, we plot the mean square error (MSE) against different values of trade-off parameters, i.e., $\psi$, $\lambda$ and $\gamma$, in the Traffic dataset. Note that the relative value of MSE is plotted to ensure the difference is distinguishable. This experiment is conducted under different settings: input-8-predict-8, input-16-predict-16, and input-32-predict-32. For $\psi$, the value ranges from 0 to 0.8, $\lambda$ ranges from 0 to 1.6, and $\gamma$ ranges from 0 to 0.5. The results are shown in Fig. 11. We can see that the model's performance varies slightly as the trade-off parameters take different values, which shows that our model is robust enough against different trade-off parameters.

### D.5 Scalability Analysis of Varying Time Series Length and Dataset Size

We additionally investigate the scalability of D$^3$VAE against different lengths of the time series and varying amounts of available data. The experiments are conducted on the Electricity and Traffic datasets, and the results are reported in Tables 8 and 9, respectively. We can observe that the predictive performance of D$^3$VAE is relatively stable under different settings. In particular, the longer the target series to predict, the worse performance might be obtained. Besides, when the amount of available data is shrunk, D$^3$VAE performs more stable than expected. Note that on the 20%-Electricity dataset, the performance of D$^3$VAE is much worse than other subsets of the Electricity dataset, mainly because the sliced 20%-Electricity dataset involves more irregular values.

## E  Disentanglement for Time Series Forecasting

Fig. 13 illustrates the disentanglement of latent variable $Z$ for time series forecasting. It is difficult to choose suitable disentanglement factors under the unsupervised learning of disentanglement.

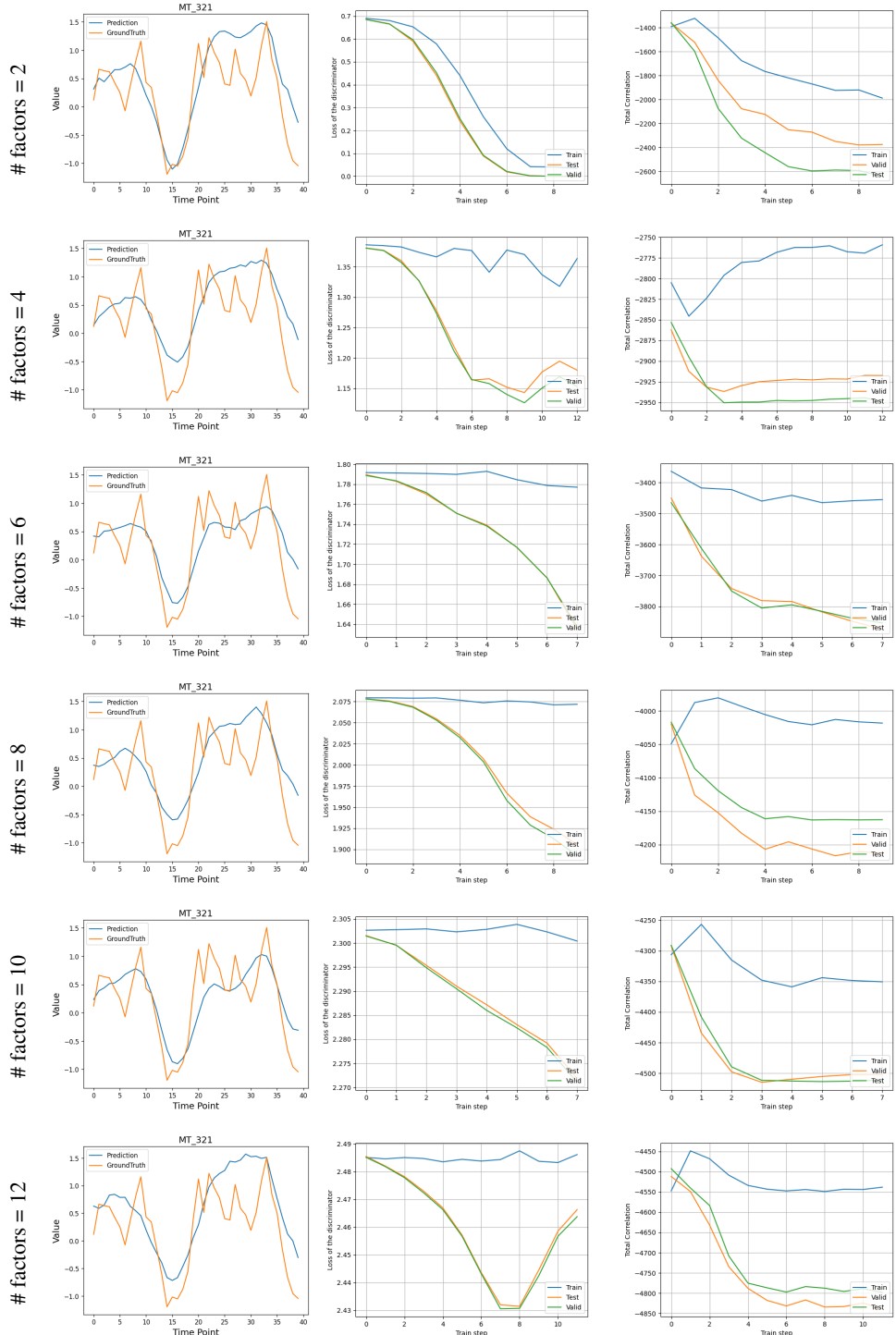

Figure 12: We showcase an instance from the Electricity dataset and demonstrate the results when different numbers of factors in disentanglement are adopted. For each row, from left to right, the prediction result of TSF, the learning curve of the discriminator, and the total correlation are plotted, respectively.

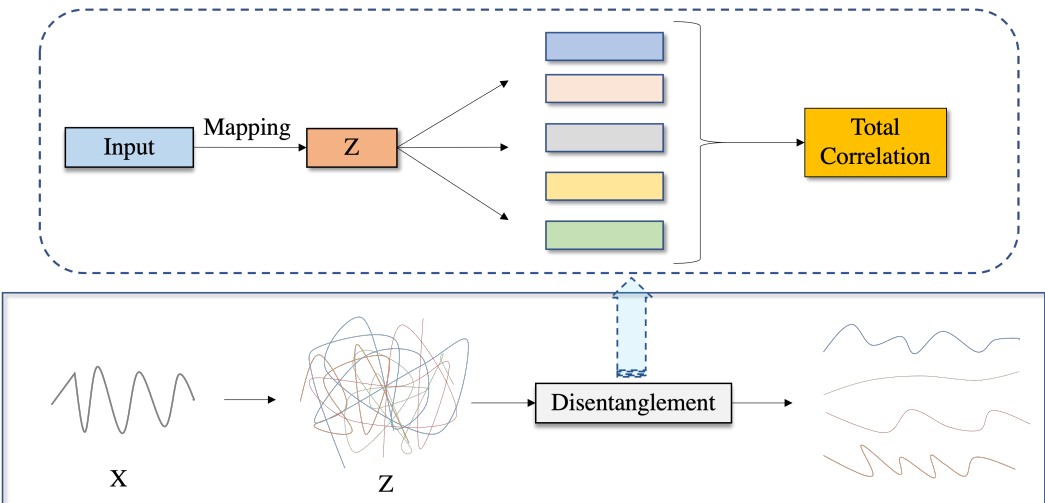

Figure 13: Disentangling latent variable $Z$ of time series. Specifically, the input $X$ is first mapped into $Z$. Then $\forall z_i \in Z$ is decomposed as $z_i = [z_{i,1}, \cdots, z_{i,m}]$ and the metric of total correlation is utilized to minimize the inter-dependencies among "hand-crafted" factors. In this way, the disentangled factors tend to be not only discriminative but also informative.

---

**Algorithm 3** Train a discriminator for time series disentanglement.

---

1: **repeat**
2:     Initialize the loss of a discriminator $\mathcal{D}_\varphi$: $L(\mathcal{D}_\varphi) = 0$
3:     Decompose the latent variable generated in Algorithm 1 as $z_i = [z_{i,1}, z_{i,2}, ..., z_{i,m}]$ $(i = 1, \cdots, n)$
4:     **for** $z_i$ in **Z do**
5:         $L = L + \sum_{j=1}^{m} \|\mathcal{D}_\varphi(z_{i,j}) - j\|^2$
6:     **end for**
7:     Optimize the discriminator: $\varphi \leftarrow \operatorname{argmin}(L)$
8: **until** Convergence

---

Therefore, we attempt to inspect the TSF performance against different numbers of factors to be disentangled. We implement a simple classifier as a discriminator to further evaluate the disentanglement quality in Fig. 12 (and Algorithm 3 demonstrates the training procedure of the discriminator). To be specific, we take different dimensions of $Z$ as the factors to be disentangled: $z_i = [z_{i,1}, \cdots, z_{i,m}]$ $(z_i \in Z)$, then an instance consisting of factor and label $(z_{i,j}, j)$ is constructed. We shuffle these $m$ examples for each $z_i$ and attempt to classify them with a discriminator, then the disentanglement can be evaluated by measuring the loss of the discriminator. The learning curve of the discriminator can be leveraged to assess the disentanglement, and the discriminator is implemented by an MLP with six nonlinear layers and 100 hidden states. The results of prediction, discriminator loss, and the total correlation w.r.t. different numbers of factors are plotted in Fig. 12, respectively. As shown in Fig. 12, the number of factors does affect the prediction performance, as well as the disentanglement quality. On the other hand, the learning curves can be converged when different factors are adopted, which validates that the disentanglement of the latent factors is of high quality.

In addition to the above method evaluating the disentanglement indirectly, we adopt another metric named Mutual Information Gap (MIG) [11] to evaluate the quality of disentanglement in a more straightforward way. Specifically, for a latent variable $z_i \in Z$, the mutual information between $z_{i,j}$, and a factor $v_k \in [1, m]$ can be calculated by

$$I_d(z_{i,j}, v_k) = \mathbb{E}_{q(z_{i,j}, v_k)}[log \sum_{d \in \mathcal{S}_{v_k}} q(z_{i,j}|d)p(d|v_k)] + H(z_{i,j}), \tag{16}$$

where $d$ denotes the sample of $z_{i,j}$ and $\mathcal{S}_{v_k}$ is the support set of $v_k$. Then, for $z_{i,j}$

$$MIG(z_{i,j}) = \frac{1}{m} \sum_{1}^{m} \frac{1}{H(v_k)}(max(I_d(z_{i,j}, v_k)) - submax(I_d(z_{i,j}, v_k))), \tag{17}$$

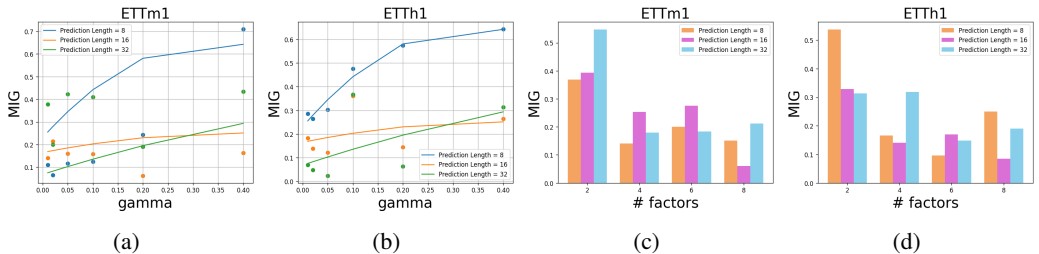

Figure 14: We evaluate the quality of disentanglement on ETTm1 and ETTh1 datasets regarding the mutual information gap (MIG). **(a-b)** The scatter plots of the MIG against varying weights $\gamma$ in loss function (refer to Eq. (14) in the main text). **(c-d)** MIG v.s. different numbers of factors.

where $submax$ means the second max value of $I_d(z_{i,j}, v_k)$, then the MIG of $Z$ can be obtained as

$$MIG(Z) = \sum_{i=1}^{n} MIG(z_i), \quad MIG(z_i) = \frac{1}{m} \sum_{j=1}^{m} MIG(z_{i,j}). \tag{18}$$

We evaluate the quality of disentanglement in terms of MIG on ETTm1 and ETTh1 datasets, respectively, which can be seen in Fig. 14. From Figs. 14a and 14b, when the weight of disentanglement (i.e., $\gamma$ in Eq. (14) of the main text) grows, the disentangled factors are of higher quality. In other words, the latent variables can be disentangled with the help of the disentanglement module in D$^3$VAE. In addition, we examine the changes in MIG against different numbers of factors. We can observe that the difficulty of disentanglement climbs up as the number of factors increases.

## F  Model Inspection: Coupled Diffusion Process

To gain more insights into the coupled diffusion process, we demonstrate how a time series can be diffused under different settings in terms of variance schedule $\boldsymbol{\beta}$ and the max number of diffusion steps $T$. The examples are illustrated in Fig. 15. It can be seen that when larger diffusion steps or a wider variance schedule is employed, the diffused series deviates far from the original data gradually, which may result in the loss of useful signals, like, temporal dependencies. Therefore, it is important to choose a suitable variance schedule and diffusion steps to ensure that the distribution space is deviated enough without losing useful signals.

## G  Necessity of Data Augmentation for Time Series Forecasting

Limited data would result in overfitting and poor performance. To demonstrate the necessity of enlarging the size of data for time series forecasting when deep models are employed, we implement a two-layer RNN and evaluate how many time points are required to ensure the generalization ability. A synthetic dataset is adopted for this demonstration.

According to [17], we generate a toy time series dataset with $n$ time points in which each point is a $d$-dimension variable:

$$w_t = 0.5w_{t-1} + \tanh(0.5w_{t-2}) + \sin(w_{t-3}) + \epsilon, \quad X = [w_1, w_2, ..., w_n] * F + v$$

where $w_t \in \mathcal{R}^2$, $F \in \mathcal{R}^{2 \times d} \sim \mathcal{U}[-1, 1]$, $\epsilon \sim \mathcal{N}(0, I)$, $v \sim \mathcal{N}(0, 0.5I)$, and $d = 5$. An input-8-predict-8 window is utilized to roll this synthetic dataset. We split this synthetic dataset into training and test sets with a ratio of 7:3. We train the RNN in 100 epochs at most, and the MSE loss of training and testing are plotted in Fig. 16. It can be seen that the inflection points of the loss curves move back gradually and disappear as increasing the size of the dataset. Besides, with fewer time points, like, 400, the model can be overfitted more easily.

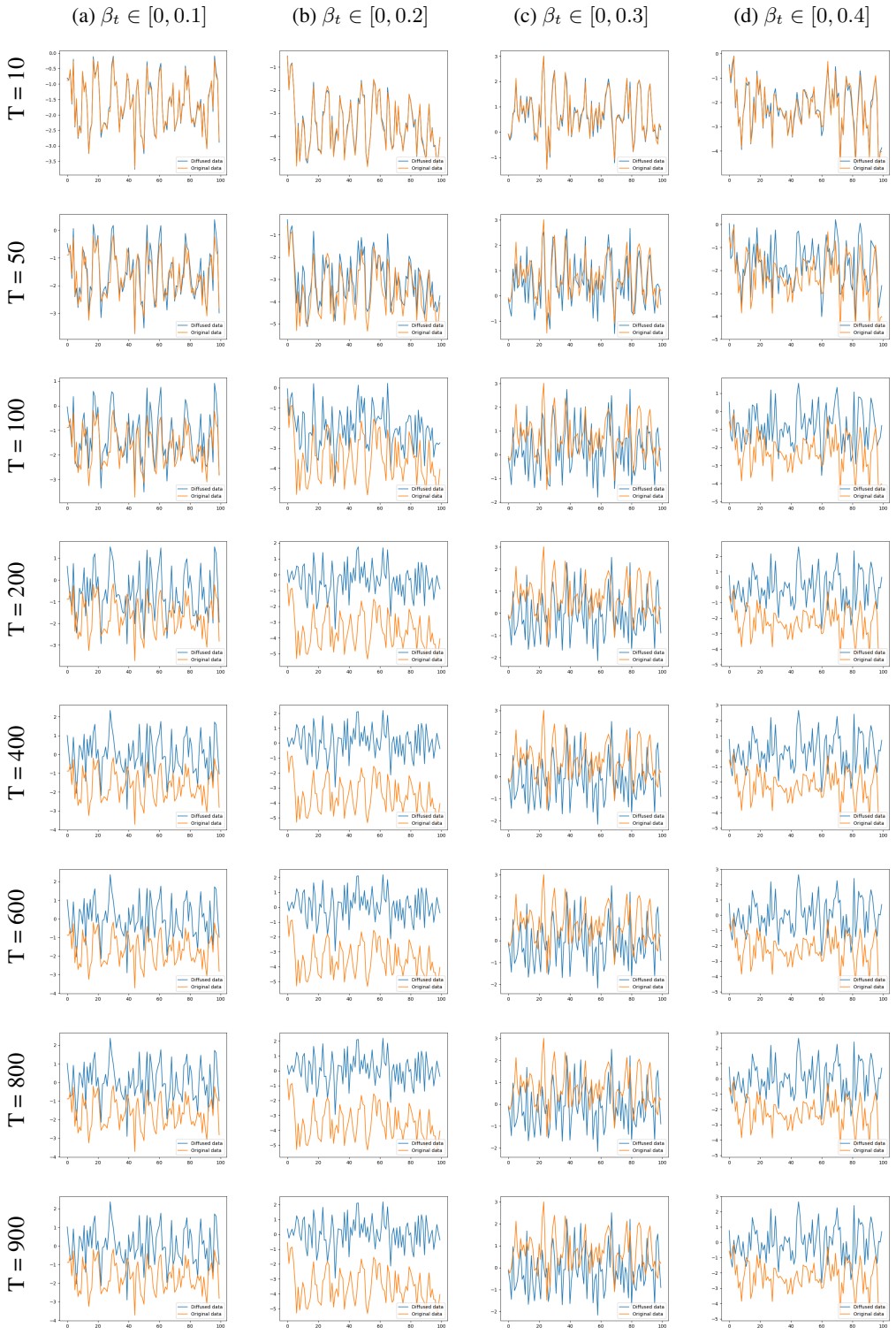

Figure 15: Diffused time series with different variance schedules and diffusion steps. We randomly choose a sample series from the synthetic dataset D2 and plot the original time series data, as well as the diffused series.

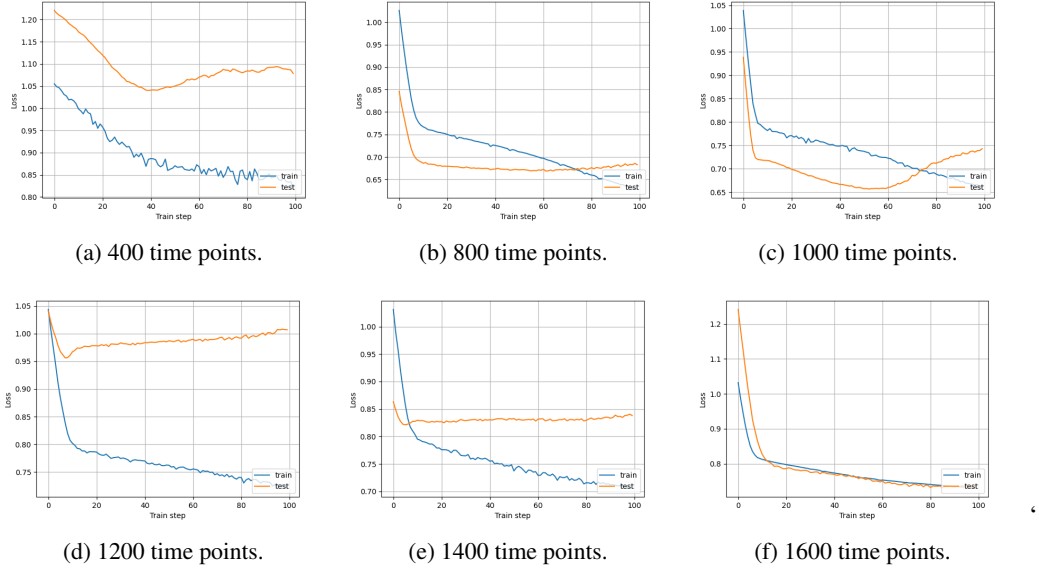

Figure 16: The curves of training and testing losses when the available time series data are of different sizes.