# OpenReview forum: "Generative Time Series Forecasting with Diffusion, Denoise, and Disentanglement"
_NeurIPS.cc/2022/Conference — NeurIPS 2022 Accept_

### Official Review · Reviewer_ee78 · 2022-07-10

**Rating:** 7
**Confidence:** 4
**Soundness:** 3 good
**Presentation:** 4 excellent
**Contribution:** 4 excellent

**Summary:**

The authors present a time-series forecasting model that can handle short examples. This is done using a generative model based on a BVAE. The experiments prove an excellent performance relative to SOTA techniques.

**Questions:**

1. How would the performance of the system change if the time series were shortened?
2. Why is the estimated noise discontinuous, when the ground truth is not.


**Limitations:**

I think this work is very useful for all those using time series forecasting for running their businesses. The beneficial impact that this technique could have on, for example, optimizing inventory levels, is very important.

**Strengths And Weaknesses:**

Strengths:
1. The presented method is developed using powerful theoretical ideas. This makes it easy to justify and understand the underlying reasons that explain the design.
2. The experiments show an important improvement relative to the specified benchmark.
Weaknesses:
1. Fig. 2 shows important discontinuities in the estimated noises. These sudden breaks of continuity are not reflected or explained by the ground truth. Hence, there is a high probability that they are caused by numerical instabilities in the numerical algorithms.
2. There is no hyperparameter search, which strongly hints that there might be versions of the same pipeline capable of reaching even better performances.

---

> ### Author Response · Authors · 2022-08-02
> **Really appreciate your positive comments on our work.**
>
>
>
> > 1. How would the performance of the system change if the time series were shortened?
>
> We have shorten the time series used in this paper, in the main text, all the time series data (including the real-world and synthetic datasets) are no more than 1000 time points. Besides, we report the evaluation results with the input and predict lengths ranging from 8 to 32 in the performance comparison experiments.
>
> Besides, we report the performance changes when the available data is shrunk, as well as the time series length, is shortened, in Appendix D.5 in the revised version for rebuttal.
>
>
> > 2. Why is the estimated noise discontinuous, when the ground truth is not.
>
> With the increase of the diffusion steps, the distribution of the time series also changes gradually, and when the diffusion steps reach a certain level, the diffused data will have a totally different distribution from the original time series which we have discussed in Section 3.3. This is why we need to control the diffusion steps and the diffusion parameters to ensure that we could enlarge the distribution space while retaining useful features.
>
> And when the prediction results have the same distribution as the ground truth (same trend or peak value), which means there is little uncertainty exists in the prediction, the estimated noise will present in a small value, but it's not discontinuous.
> When the prediction results show a clear difference (peak delay or opposite tendency) the estimated noise will keep in a bigger range. This property of the estimated noise could reflect the uncertainty of the prediction properly.

---

### Official Review · Reviewer_4VLY · 2022-07-11

**Rating:** 5
**Confidence:** 4
**Soundness:** 3 good
**Presentation:** 2 fair
**Contribution:** 3 good

**Summary:**

The author proposed D3VAE, a coupled diffusion probabilistic model for time series forecasting problems. A diffusion process is introduced on top of the BVAE (Bidirectional Variational Auto-Encoder) to mitigate the effect of the random noise. Total correlation is used to disentangle the different latent variables Z to better extract dependencies from the input data hence improving the overall performance. Last, a denoising score matching (DSM) module is used to improve the accuracy of the generated results. Experimental results show that D3VAE perform better than other baseline models in most of the synthetic and real-world datasets.

**Questions:**

1. Which evidence can prove the learned latent variables Z are actually disentangled?
2. Which part of the results shows that the interpretability is improved compared to other baseline methods?
3. What is the actual architecture of D3VAE?

**Limitations:**

Improvement:
1. Add experiments for evaluating the disentanglement of the learned representations.
2. Add graphs to illustrate the increase in interpretability.
3. Refine the writing.

**Strengths And Weaknesses:**

Pros:
1. The design of this model is quite ingenious. The introduction of the DRL (disentangled representation learning) could cause noises in the reconstruction; therefore, a denoising module is used for better prediction accuracy.
2. An extensive amount of experimental results are provided regarding the effectiveness of each module over a large scale of synthetic and real-world datasets.
3. Experimental results show that D3VAE perform better than other baseline models in most of the synthetic and real-world dataset.


Cons:
1. There are not enough results for the disentanglement section. The author claim that using the TC can force the model to disentangle the latent variables {z1, z2, ..., zn}. However, there is not enough evaluation or results to prove that the learned latent variables are actually disentangled. The supplementary document only includes a graph that shows the difference between the ground truth and the prediction with different numbers of z used. That does not prove the claim that the author makes. The actual level of disentanglement for each zi or the single variables in zi are not investigated properly.
2. The overall structure and procedures of the proposed D3VAE are vague since there is only an illustration of the architecture of BVAE. It is hard for me to picture each module and its locations in the whole pipeline.
3. The claim of interpretability is not addressed in the paper. In section 2.4, the author states that the introduction of disentanglement can lead to better interpretability. However, there is not enough evidence regarding this topic. Although there are case studies in the supplementary document which might shed some insights, the relationship between those insights and the usage of disentangled representation learning is missing.

---

> ### Author Response · Authors · 2022-08-02
> **Many thanks for your positive reviews about our work and constructive comments of the disentanglement part.**
>
> > Which evidence can prove the learned latent variables Z are actually disentangled?
>
> In the time-series-related tasks, it is challenging to visualize the latent variables in a more human-readable and explainable way. To assess the disentanglement quality of the latent variables, we reformulate the problem of "Whether the latent variables $z_{1:n}$ can be fully disentangled?" into another problem "Whether the factors of each $z_i$ can be classified correctly?". Therefore, we learn a discriminator to determine whether a factor of latent variable $z_i$ belongs to the corresponding category (i.e., $z_{i,j}$ belongs to class j). If the classifier can discriminate the factors of each latent variable in $(z_1, z_2, \cdots, z_n)$ perfectly, the latent variable Z can be well disentangled. We have revised the analysis of the results reported in Appendix C accordingly.
>
> In addition, we adopt the Mutual Information Gap (MIG) as a straightforward metric to evaluate the disentanglement quality, which has also been reported in Appendix C in the revised paper for rebuttal. With the experimental results in terms of MIG, we can validate that the latent variables are indeed disentangled.
>
>
> > ... Although there are case studies in the supplementary document which might shed some insights, the relationship between those insights and the usage of disentangled representation learning is missing.
>
> Thanks for pointing out this issue. How to leverage the disentangled representations to deliver insights for time series forecasting (esp. high dimensional time series) is really a big challenge and we will continue to devote ourselves to this topic in the future.
>
>
> > Which part of the results shows that the interpretability is improved compared to other baseline methods?
>
> We predict the distribution of the time series, so the variance of the prediction is delivered, and by controlling the diffusion level according to different types of time series, the variance is fit into a reasonable range, by which we can estimate the uncertainty of the prediction accordingly, and thus provide more reliability and interpretability for down-stream tasks.
>
> Most of the baseline methods we compared in this work don't involve the interpretability part, so it is hard to compare the interpretability of our method to the other baselines.
>
> Besides, as for our method, the disentanglement of latent variables could improve the interpretability of the model [1, 2], also the combination of the disentanglement and the BVAE could ensure the effectiveness of the disentanglement.
>
>
> [1] Li Y, Chen Z, Zha D, et al. Interpretable time-series representation learning with multi-level disentanglement[J]. arXiv preprint arXiv, 2021, 2105.
>
> [2] Hsu W N, Zhang Y, Glass J. Unsupervised learning of disentangled and interpretable representations from sequential data[J]. Advances in neural information processing systems, 2017, 30.
>
>
> > What is the actual architecture of D3VAE?
>
> We have added the picture of the global architecture of $D^{3}VAE$ in the revised paper for rebuttal.

---

### Official Review · Reviewer_GcMT · 2022-07-12

**Rating:** 5
**Confidence:** 3
**Soundness:** 3 good
**Presentation:** 3 good
**Contribution:** 3 good

**Summary:**

In this submission, the authors propose to use coupled diffusion model with denoising mechanism to augment the time series. The forecasting can be performed directly with by the generative model. In particular, the diffusion model aims to deal to aleatoric uncertainty. Also, the authors propose to perform latent variable disentangling to enhance interpretation.

**Questions:**

1. What is the actual relationship between the aleatoric uncertainty and the diffusion model?
2. What is the advance of the diffusion model in this work wrt the application to time series?
2. What is the connection between the diffusion model and disentanglement?

**Limitations:**

The authors have mentioned several limitations in the main text. I agree with the authors that the process is sensitive to hyperparameters such as beta and time steps. One missing part is the computation performance, which is a common concern when using diffusion model.

**Strengths And Weaknesses:**

Originality
This method is built on several original insights, especially on the connection between noise and diffusion model. On the other hand, the model components each as BVAE and denoising disentangling are extended from previous works.

Quality:
1.	The conductance of this study is of high quality in general. The experiments are solid with several ablations.
2.	However, one missing part in the discussion of mythological advance (not difference) with previous generative or diffusion model. For example, TimeGrad also relies on the diffusion model. Why this method outperforms a lot in the experiments.
3.	The overall presentation is satisfactory. However, it would be more reader-friendly for the audience to follow if there is a global picture in the main text. Also, several examples can be moved from the appendix to the main text.

Clarity：
1.	The motivation of the method is clearly addressed. The author emphasizes the noise challenges in short time forecasting setting.
2.	As the uncertainty itself is analyzed by separating aleatoric and epistemic uncertainty, the authors state that their proposed method can reduce the former. However, the reviewer finds it hard to follow the derivation of this conclusion (i.e., Lemma 1-2). It seems no bounds of this uncertainty are given.
3.	The connection between the diffusion model and disentanglement is missing? Why are these two components combined in one work?

Significance
This paper tackles an important issue and highlights a feasible augmentation strategy for noisy input.

---

> ### Author Response · Authors · 2022-08-02
> **Many thanks for your detailed and insightful comments. We really appreciate your constructive suggestions and have revised the rebuttal version of our submission accordingly. In particular, we provide a global picture of our proposed method and differentiate our proposed diffusion model from previous works by describing the devised BVAE in more detail.**
>
> > one missing part in the discussion of mythological advance...
>
> Our proposed diffusion model makes advances compared to existing diffusion models in three aspects:
> 1) The diffusion model is more flexible than other generative models. To explore such expressiveness for time series forecasting, in our proposed diffusion model, the forward diffusion process is applied to both input series and output series synchronously. Then, the downstream VAE model can be empowered a lot.
> 2) For inference, a BVAE model is developed to take the place of the reverse process in the traditional diffusion model. Existing diffusion models aim to generate higher quality samples. In the time series forecasting task, more accurate prediction is expected. To achieve this, we substitute the reverse process for a VAE to implement the inference process which is more tractable and efficient compared to existing diffusion models. Besides, the gap between the input and output can be well captured by the VAE model.
> 3) The disentanglement is employed thanks to the structure of BVAE such that the dimensions of the latents can be treated as the factors to be disentangled and a more reliable prediction can be achieved.
>
> In particular, TimeGrad makes the prediction in an autoregressive way, it learns the distribution of input series on top of the diffusion model and the RNN network. Then, it infers the target series with the learned distribution of the input series. Hence, the generalizability of TimeGrad for future prediction is limited due to the gap between the input series and the output series. In addition, under the short time series forecasting settings, the RNN can not learn informative signals effectively.
>
>
>
> > Hard to follow Lemma 1-2
>
> We have rephrased Lemma 2 in the main text of the rebuttal version and added more detailed derivations for Lemma 1 and Lemma 2 in Appendix H.
>
> > no bounds of this uncertainty
>
> For the bounds of the uncertainty, the estimation of the uncertainty in Eq. (11) in the main text, i.e.,  $\sigma_0^2 \nabla_{\widehat{Y}} E (\widehat{Y}; \zeta)$, can be regarded as the upper bound of the aleatoric uncertainty. More dedicated bound analysis of the aleatoric uncertainty will be provided in the future.
>
>
> >connection between the diffusion model and disentanglement
>
> The diffusion model and the disentanglement are connected via the BVAE. BVAE acts as the inference process of the diffusion model and the multiple latent variables yielded by the BVAE can be leveraged by the disentanglement.
>
>
> >the computation performance
>
> Thanks for pointing out this issue, the computational complexity analysis is as below and we will add a more comprehensive analysis in the future.
>
> In this work, the (forward) diffusion process is determined by the variance schedule which is fixed to constants, and for the (reverse) inference process, a BVAE model equipped with single step gradient denoising jump is developed, which is more efficient than the traditional diffusion model.
>
>
> >What is the actual relationship between the aleatoric uncertainty and the diffusion model?
>
> We use the diffusion process to perturb the distribution space of the original time series, and by diffusing the input series and the target series synchronously (the noise level for the input series is larger than the target series), the aleatoric uncertainty contained in the time series is diffused as well. Besides, with the number of diffusion steps and the noise level properly configured, the introduced noise, as well as the aleatoric uncertainty, will not climb up (which is guaranteed by Lemma 2 in Section 2.2). Afterward, a BVAE is developed to take the place of the reverse diffusion process of the traditional diffusion model, such that no diffusion process is involved in the inference. Moreover, with the denoising score matching module, the aleatoric uncertainty will be further reduced.
>
>
> >What is the advance of the diffusion model in this work wrt the application to time series?
>
> With the proposed diffusion model, the time series data is augmented without increasing the aleatoric uncertainty such that the generalization ability of the deep model can be enhanced. Besides, by replacing the reverse diffusion process with BVAE, the generative model becomes more tractable for time series forecasting. Furthermore, more reliable predictions can be delivered by providing uncertainty estimation captured by the denoising score matching module.
>
>
> >What is the connection between the diffusion model and disentanglement?
>
> In this work, the reverse process of the diffusion model is implemented by a bidirectional variational auto-encoder (BVAE). The side-output of BVAE offers us an opportunity to treat the latent variables in a multivariate fashion and exploit the informative latent variables such that the disentanglement can be leveraged to examine the factors for each $z_i$ in $z_{1:n}$, which is helpful in improving the interpretability for time series forecasting.

---

### Meta-Review · Area_Chair_pLFU · 2022-08-29

**Recommendation:** Accept
**Confidence:** Certain

**Metareview:**

I recommend to accept this paper.

In this paper, the authors propose to address the time series forecasting problem with a generative modeling technique called D3VAE with both theoretically sounded insights and extensive empirical results. All the reviewers agree to accept this paper. I would encourage the authors to revise the paper based on the suggestions provided by reviewers.

**Award:**

No

---

### Decision · Program_Chairs · 2022-09-14

Accept